# Genomic epidemiology of SARS-CoV-2 reveals multiple lineages and early spread of SARS-CoV-2 infections in Lombardy, Italy

Claudia Alteri [1,11], Valeria Cento[1,11], Antonio Piralla [2,11], Valentino Costabile [3], Monica Tallarita[2], Luna Colagrossi[4], Silvia Renica[1], Federica Giardina[2], Federica Novazzi[2], Stefano Gaiarsa[2], Elisa Matarazzo[5], Maria Antonello[1], Chiara Vismara[6], Roberto Fumagalli[7], Oscar Massimiliano Epis[8], Massimo Puoti[9], Carlo Federico Perno [4✉] & Fausto Baldanti[2,10]

From February to April 2020, Lombardy (Italy) reported the highest numbers of SARS-CoV-2 cases worldwide. By analyzing 346 whole SARS-CoV-2 genomes, we demonstrate the presence of seven viral lineages in Lombardy, frequently sustained by local transmission chains and at least two likely to have originated in Italy. Six single nucleotide polymorphisms (five of them non-synonymous) characterized the SARS-CoV-2 sequences, none of them affecting N-glycosylation sites. The seven lineages, and the presence of local transmission clusters within three of them, revealed that sustained community transmission was underway before the first COVID-19 case had been detected in Lombardy.

[1] Department of Oncology and Hemato-oncology, University of Milan, Milan, Italy. [2] Molecular Virology Unit, Microbiology and Virology Department, Fondazione IRCCS Policlinico San Matteo, Pavia, Italy. [3] Department of Pathophysiology and Transplantation, University of Milan, Milan, Italy. [4] Microbiology and Diagnostic Immunology Unit, Bambino Gesù Children's Hospital, IRCCS, Rome, Italy. [5] Residency in Microbiology and Virology, University of Milan, Milan, Italy. [6] Chemico-clinical and Microbiological Analyses, ASST Grande Ospedale Metropolitano Niguarda, Milan, Italy. [7] Department of Anesthesiology, Critical Care and Pain Medicine, ASST Grande Ospedale Metropolitano Niguarda, Milan, Italy. [8] Rheumatology Unit, ASST Grande Ospedale Metropolitano Niguarda, Milan, Italy. [9] Infectious Diseases Unit, ASST Grande Ospedale Metropolitano Niguarda, Milan, Italy. [10] Department of Clinical, Surgical, Diagnostic and Paediatric Sciences, University of Pavia, Pavia, Italy. [11] These authors contributed equally: Claudia Alteri, Valeria Cento, Antonio Piralla. ✉email: cf.perno@uniroma2.it

Since coronavirus disease 2019 (COVID-19) was initially reported in China on 30 December 2019 (refs. [1,2]), SARS-CoV-2 has been spreading worldwide and, as of June 14, 2020, there have been 7.55 million confirmed infections and 423,000 deaths reported worldwide (World Health Organization, 2020).

In Italy, the first case of evident SARS-CoV-2 transmission emerged on February 20, in Codogno, Lombardy, when a young man affected by interstitial pneumonia was diagnosed for SARS-CoV-2. From that date, the number of diagnosed COVID-19 cases exponentially increased, and Lombardy became the area most affected by the COVID-19 pandemic, totaling 89,018 infections by June 2, out of a total of 233,197 cases in Italy. Thereafter, the COVID-19 epidemic grew exponentially during the first days of March 2020, peaking on March 21, 2020, with 6557 newly confirmed cases. Two months later, reported COVID-19 cases in Italy dropped to ~600 per day, indicating the first wave of epidemic was nearing containment.

With a population of 10 million, Lombardy is the most densely populated region in Italy, and one of the largest. Milan is the largest metropolitan area in Italy and the third most densely populated functional urban area in Europe[3], with well-established economical and transportation links to Europe and beyond. This scenario creates the conditions to host and favor the spread of a highly transmissible virus such as SARS-CoV-2.

In this study, an integrated approach comprising epidemiological and viral genetic data was used to reconstruct the pattern of SARS-CoV-2 spread in this region. Whole-genome sequencing was performed for 346 SARS-CoV-2 strains obtained from individuals from various geographical areas in a time span of 2 months. The main aim of this study was to trace local transmission chains and the temporal and geographical evolution of the virus in Lombardy in relation to epidemiological data and measures implemented to contain the outbreak.

## Results

**Patient characteristics**. From February 22 through April 4, 2020, nasopharyngeal swabs taken from a total of 25,082 patients were screened for SARS-CoV-2 infection at two major hospitals in Lombardy (Supplementary Fig. 1). A diagnosis of COVID-19 was made for 11,445 of them. Whole-genome sequencing was performed in 371 samples collected from 371 patients residing in all 12 provinces of Lombardy and with varying disease symptoms ranging from mild to critical. Sampling selection criteria for these samples and the comparison of their demographic and clinical characteristics with SARS-CoV-2-infected general population are illustrated in Supplementary Fig. 1, Supplementary Table 1, and Supplementary Results.

Twenty-five samples were excluded due to failed amplification ($n = 9$) or poor genomic coverage (<60%, $n = 16$). The final study population thus consisted of 346 patients, whose geographical, demographic, and clinical characteristics are reported in Fig. 1 and Table 1.

One-hundred and ninety-five (56.4%) were male, and the median age was 72 (IQR: 54–84) years. Fever was the most common COVID-19 symptom at admission, followed by cough and dyspnea. Chest radiographs or CT scan confirmed a classical bilateral interstitial pneumonia for 49.6% of them. Patients with critical or severe COVID-19 manifestation more frequently suffered from at least one chronic comorbidity, compared to those with moderate and mild manifestations ($P = 0.002$ by Chi-square test for trend), with greater impact played by hypertension and chronic kidney disease ($P < 0.001$, respectively by Chi-square test for trend). Regarding contact tracing, 25 reported to be COVID-19 contact while only four patients out of 237 declared an international travel history. For three of them, the destination

was not specified, while one individual reported travel in Singapore from January 12 to February 5, 1 month before receiving a SARS-CoV-2 diagnosis of a B.1.1 strain infection (March 10).

**Genome coverage**. SARS-CoV-2 sequence reads were able to cover from 94.0% to 99.7% of the reference genome (GenBank: NC_045512.2), independently of SARS-CoV-2 load (Supplementary Fig. 2). The few genome regions ($N = 4$) with lower reads coverage were consistently limited to no more than 35 nt.

**Lineages and single nucleotide polymorphisms characterizing SARS-CoV-2 sequences**. By reassigning sequences to the lineages according to the PANGOLIN application[4], we found that the majority of sequences ($n = 179$, 51.7%) belonged to lineage B.1, 126 (36.4%) belonged to lineage B.1.1, and 34 (9.8%) were assigned to lineage B.1.5. Lineage B.1.1.1 was found in four north Italian sequences, while A.2, B, and B.1.8 lineages were detected in one patient each (Fig. 2). Demographic and clinical characteristics of patients infected with SARS-CoV-2 from such lineages are reported in Table 1. While no relevant differences in demographic and clinical characteristics were found among lineages, some differences were found in the geographical distribution of the three most represented lineages in Lombardy (B.1, B.1.1, and B.1.5). Lineage B.1 mostly affected southern Lombardy, including Lodi, Cremona, and Mantua, with 90.2% of these provinces' sequences belonging to this lineage (55/61, Fig. 1 and Table 1), while in the north of Lombardy, mostly in Como, Lecco, Bergamo, and its adjacent territories such as Alzano and Nembro, 62.7% of sequences belonged to B.1.1 lineage (84/134, Fig. 1 and Table 1). Finally, lineage B.1.5 was mainly detected in Pavia, with a local cluster in this area composed of 23 B.1.5 sequences.

The composition of sequences did not change substantially from the early testing phase (before lockdown, March 8) to later time points (after March 8), even if B.1.1 lineage, which had made up 27 out of 106 sequences (25.5%) in the early phase of testing, increased to 99 out of 240 haplotypes (41.2%) in the later phases of testing, mostly because of the decreased frequency of the B.1 and other B.1 derived lineages (Table 1 and Fig. 3a). Looking at the genetic distance with respect to SARS-CoV-2 reference strain (belonging to lineage B), it was naturally lower for isolates in lineage B.1 with respect to isolates in lineage B.1.1 and B.1.5 ($1.4 \times 10^{-4}$ [$1.0 \times 10^{-4}$; $1.7 \times 10^{-4}$] vs $1.7 \times 10^{-4}$ [$1.4 \times 10^{-4}$; $1.7 \times 10^{-4}$] vs $2.4 \times 10^{-4}$ [$2.1 \times 10^{-4}$; $2.7 \times 10^{-4}$], $P < 0.001$ by Kruskal–Wallis test, Fig. 3b), thus indicating a closer genetic relatedness to original B strains for lineage B.1 with respect to the others. Concordantly with the increase of lineage B.1.1 over time, the genetic pairwise distance of the 346 sequences indicated that the SARS-CoV-2 sequences evolved progressively during time (rho = 0.585 and 0.582 respectively, Fig. 3c and Supplementary Fig. 3).

Six single nucleotide polymorphisms (SNPs) were shared among >10 north Italian genomes, with a prevalence ranging from 3.2% (20268, A to G, syn in nsp15) to 99.4% of the SNP 23403, A to G, non-syn D to G in S and 14408, C to T, non-syn P to L in RdRp (Fig. 4a, b). All SNPs were detected in at least one previously published SARS-CoV-2 sequence in GISAID. Five out of these six SNPs were non-synonymous, and three were mapped within SARS-CoV-2 structural proteins. Notably, two non-synonymous SNPs reside in S protein: the C to T at position 23575 (S protein; amino acid T to I; intra-patient prevalence ~100%) found in 24 north Italian sequences and previously detected in <1% of samples in China, the rest of Europe and North America, and the A to G at position 23403 (S protein;

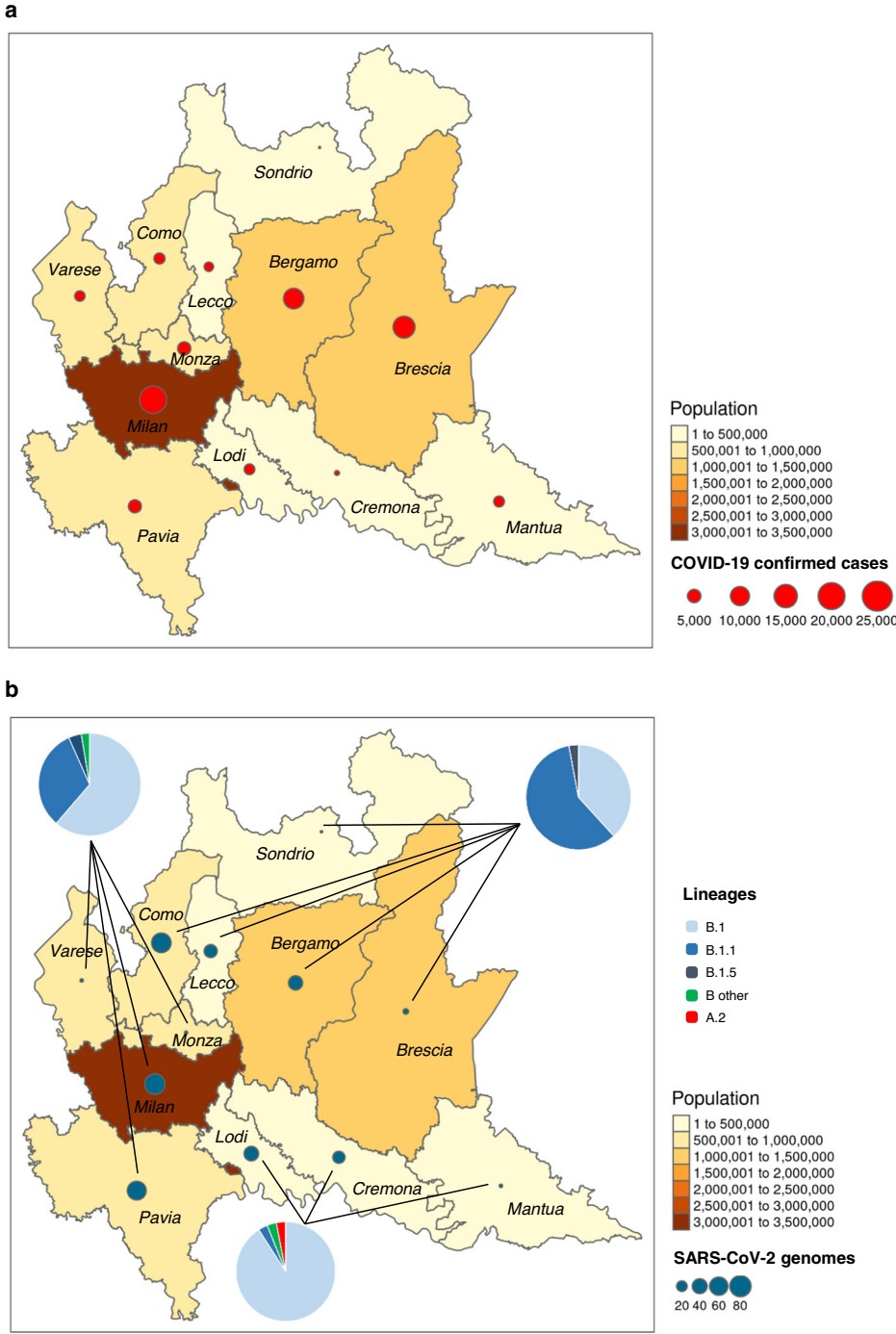

**Fig. 1 Distribution of SARS-CoV-2 in Lombardy. a** Geographic distribution of COVID-19 confirmed cases and population density among the 12 provinces of Lombardy. **b** Geographic distribution of SARS-CoV-2 genomes, the lineages detected, and population density among the 12 provinces of Lombardy. Maps were obtained, thanks to the free available software Rgui using the R statistic library tmap.

amino acid D to G; intra-patient prevalence ~54%), present in 344 north Italian sequences and firstly detected in China with a prevalence of 1.7%, then rapidly selected and spreading in all countries with a prevalence ranging from ~17% in Asia to ~70% in Europe and North America.

**Emergence of lineages in Lombardy**. The Bayesian molecular clock analysis estimated a mean evolutionary rate of $1.16 \times 10^{-3}$ subs/site/year (95% HPD, $1.01 \times 10^{-3}$–$1.32 \times 10^{-3}$).

The current low genetic diversity of SARS-CoV-2 genomes worldwide implies very low posterior probabilities (<0.50) for most of the nodes. Nevertheless, it was possible to identify clear transmission chains and clusters with posterior probability support ≥0.98, able to clarify the dynamic of viral lineages in Lombardy (Fig. 5, Table 2).

As stated above, two sequences from Lombardy clustered with A.2 and B sequences, respectively (Fig. 2). The patient infected by lineage A.2 was diagnosed on March 15 in Lodi. Unfortunately, travel information and possible COVID-19 contact were

**Table 1 Demographic and clinical findings of the 346 SARS-CoV-2-infected patients according to lineages.**

| | Overall, N = 346 | Lineages | | | | P-value[b] |
|---|---|---|---|---|---|---|
| | | B.1, N = 179 | B.1.1, N = 126 | B.1.5, N = 34 | Other[a], N = 7 | |
| **Demographics and clinical characteristics** | | | | | | |
| Age, years | 72 (54–84) | 68 (54–83) | 74 (55–84) | 74 (60–85) | 72 (46–84) | 0.815 |
| 18–39 | 30 (8.7) | 17 (9.5) | 9 (7.1) | 4 (11.8) | 0 (0.0) | 0.484 |
| 40–49 | 36 (10.4) | 20 (11.2) | 11 (8.7) | 3 (8.8) | 2 (28.6) | 0.374 |
| 50–59 | 48 (13.9) | 31 (17.3) | 16 (12.7) | 1 (2.9) | 0 (0.0) | 0.082 |
| 60–69 | 46 (13.3) | 23 (12.8) | 16 (12.7) | 6 (17.6) | 1 (2.2) | 0.915 |
| 70–79 | 64 (18.5) | 24 (13.4) | 30 (23.8) | 8 (23.5) | 2 (28.6) | 0.101 |
| ≥80 | 122 (35.3) | 64 (35.8) | 44 (34.9) | 12 (35.3) | 2 (28.6) | 0.978 |
| Sex, male | 195 (56.4) | 87 (48.6) | 80 (63.5) | 23 (67.6) | 5 (71.4) | 0.018 |
| **Residency** | | | | | | |
| Milan | 75 (21.7) | 46 (25.7) | 24 (19.0) | 3 (8.8) | 2 (28.6) | 0.108 |
| Como | 68 (19.7) | 26 (14.5) | 40 (31.7) | 2 (5.9) | 0 (0.0) | <0.001 |
| Pavia | 61 (17.6) | 20 (11.2) | 15 (11.9) | 23 (67.6) | 3 (42.9) | <0.001 |
| Bergamo | 36 (10.4) | 12 (6.7) | 23 (18.3) | 1 (2.9) | 0 (0.0) | 0.003 |
| Lecco | 30 (8.7) | 9 (5.0) | 21 (16.7) | 0 (0.0) | 0 (0.0) | 0.001 |
| Lodi | 33 (9.5) | 30 (16.8) | 1 (0.8) | 0 (0.0) | 2 (28.6) | <0.001 |
| Cremona | 26 (7.5) | 23 (12.8) | 0 (0.0) | 3 (8.8) | 0 (0.0) | <0.001 |
| Other[c] | 17 (4.9) | 13 (7.3) | 2 (1.6) | 2 (5.9) | 0 (0.0) | 0.135 |
| **Chronic comorbidities[d]** | 155 (52.4) | 80 (51.3) | 57 (53.3) | 14 (46.7) | 4 (66.7) | 0.759 |
| Hypertension | 107 (35.8) | 52 (33.3) | 42 (39.3) | 11 (36.7) | 2 (33.3) | 0.530 |
| Obesity | 23 (7.8) | 9 (5.8) | 11 (10.3) | 2 (6.7) | 1 (16.7) | 0.477 |
| Diabetes | 31 (10.5) | 14 (9.0) | 14 (13.1) | 2 (6.7) | 1 (16.7) | 0.591 |
| Cardiovascular disease | 98 (33.1) | 46 (29.5) | 40 (37.4) | 10 (33.3) | 2 (33.3) | 0.635 |
| Chronic obstructive lung disease | 42 (14.2) | 16 (10.3) | 19 (17.8) | 5 (16.7) | 2 (33.3) | 0.175 |
| Malignancies | 35 (11.8) | 21 (13.5) | 11 (10.3) | 2 (6.7) | 1 (16.7) | 0.639 |
| Chronic kidney disease | 23 (7.8) | 8 (5.1) | 9 (8.4) | 3 (10.0) | 3 (50.0) | 0.868 |
| Chronic liver disease | 4 (1.4) | 3 (1.9) | 1 (0.9) | 0 (0.0) | 0 (0.0) | 0.788 |
| Other[e] | 28 (9.7) | 15 (9.7) | 9 (8.7) | 4 (13.3) | 0 (0.0) | 0.765 |
| **Symptoms at admission[f]** | | | | | | |
| Fever | 150 (65.2) | 73 (59.3) | 52 (68.4) | 19 (67.9) | 6 (100.0) | 0.174 |
| Cough | 106 (46.1) | 51 (41.5) | 35 (46.1) | 15 (53.6) | 5 (83.3) | 0.204 |
| Dyspnea | 93 (40.4) | 44 (35.8) | 32 (42.1) | 14 (50.0) | 3 (50.0) | 0.571 |
| Time from symptoms-onset to SARS-CoV-2 diagnosis, weeks | 0.29 (0.14–0.57) | 0.29 (0.14–0.50) | 0.33 (0.17–0.69) | 0.26 (0.00–0.37) | 0.17 (0.00–0.57) | 0.244 |
| SARS-CoV-2 diagnosis (month, day) | March, 14 March, 06–March, 20 | March, 11 March, 05–March, 21 | March, 15 March, 09–March, 20 | March, 10 March, 05–March, 20 | March, 18 March, 10–March, 23 | 0.138 |
| Before March, 8 | 106 (30.6) | 63 (35.2) | 27 (21.4) | 15 (44.1) | 1 (14.3) | 0.987 |
| After March, 8 | 240 (69.4) | 116 (64.8) | 99 (78.6) | 19 (54.9) | 6 (85.7) | |
| **Disease severity[g]** | | | | | | |
| Mild | 119 (50.9) | 71 (57.3) | 37 (47.4) | 12 (41.4) | 2 (33.3) | 0.314 |
| Moderate | 52 (22.2) | 30 (24.2) | 14 (17.9) | 7 (24.1) | 1 (16.7) | 0.732 |
| Severe | 53 (22.6) | 19 (15.3) | 22 (28.2) | 9 (31.0) | 3 (50.0) | 0.117 |
| Critical | 10 (4.3) | 4 (3.2) | 5 (6.4) | 1 (3.4) | 0 (0.0) | 0.674 |
| Evidence of Interstitial Pneumonia[h] | 116 (49.6) | 54 (43.5) | 41 (52.6) | 17 (58.6) | 4 (66.7) | 0.372 |
| **Contact tracing** | | | | | | |
| COVID-19 contact[i] | 25 (21.0) | 13 (22.0) | 8 (15.7) | 3 (42.9) | 1 (50.0) | 0.269 |
| International travel[j] | 4 (1.7) | 2 (1.8) | 2 (2.2) | 0 (0.0) | 0 (0.0) | 0.898 |
| **SARS-CoV-2 rtPCR** | | | | | | |
| Mean cycle thresholds[k] | 18.8 (16.8–20.1) | 19.1 (17.2–20.3) | 18.1 (16.5–19.9) | 19.2 (17.8–20.6) | 19.0 (16.3–21.3) | 0.050 |
| **Local clusters** | | | | | | |
| Sequences in cluster | 60 (17.3) | 25 (14.0) | 12 (9.5) | 23 (67.6) | 0 (0.0) | <0.001 |
| Cluster A | 25 (7.2) | 25 (14.0) | 0 (0.0) | 0 (0.0) | 0 (0.0) | – |
| Cluster B | 23 (6.6) | 0 (0.0) | 0 (0.0) | 23 (67.6) | 0 (0.0) | – |
| Cluster C | 6 (1.7) | 0 (0.0) | 6 (4.8) | 0 (0.0) | 0 (0.0) | – |
| Cluster D | 6 (1.7) | 0 (0.0) | 6 (4.8) | 0 (0.0) | 0 (0.0) | – |
| **SNP in SARS-CoV-2 genome** | | | | | | |
| 14408, C to T, non-syn P to L (RdRp) | 344 (99.4) | 179 (100.0) | 126 (100.0) | 34 (100.0) | 5 (71.4) | 0.002 |
| 20268, A to G, syn (nsp15) | 11 (3.2) | 0 (0.0) | 0 (0.0) | 11 (32.3) | 0 (0.0) | <0.001 |
| 23403, A to G, non-syn D to G (S) | 344 (99.4) | 179 (100.0) | 126 (100.0) | 34 (100.0) | 5 (71.4) | 0.002 |
| 23575, C to T, non-syn T to I (S) | 24 (6.9) | 1 (0.6) | 0 (0.0) | 23 (67.6) | 0 (0.0) | <0.001 |
| 26530, A to G, non-syn D to G (M) | 25 (7.2) | 25 (14.0) | 0 (0.0) | 0 (0.0) | 0 (0.0) | <0.001 |
| 28881-28883, GGG to AAC, non-syn RG to KR (N) | 130 (37.6) | 0 (0.0) | 126 (100.0) | 0 (0.0) | 4 (57.1) | <0.001 |

Data are expressed as median (IQR), or N (%).
[a]Including A.2 (n = 1), B (n = 1), B.1.1.1 (n = 4), B.1.8 (n = 1).
[b]Two-sided P-values were calculated by Kruskal–Wallis test, or Chi-square test for trend, as appropriate.
[c]Other includes Brescia, Mantua, Monza and Brianza, Sondrio and Varese.
[d]Data available for 309 patients.
[e]Including: Crohn's disease (n = 1), Hashimoto's thyroiditis (n = 3), familial lipid disorders (n = 9), rheumatoid arthritis (n = 3), amyotrophic lateral sclerosis (n = 1), cognitive disorders (n = 11).
[f]Data available for 180 patients.
[g]Data available for 237 patients.
[h]Diagnosed by X Ray or CT Scan. Data available for 232 patients.
[i]Data available for 119 patients.
[j]Data available for 231 patients.
[k]Real-time reverse transcription PCR Ct (cycle threshold) values ranged from 9 to 35 (GeneFinderTM COVID-19 Plus RealAmp Kit, ELITech; AllplexTM 2019-nCoV Assay, Seegene; Corman et al.[46]).

unavailable for this patient, thus hampering the tracing of the exact dynamic of the infection. Looking at the topology of the trees, the European transmission chain probably originated in Spain (Fig. 5 and Supplementary Fig. 4) and continued in France in the second half of February. The patient infected by lineage B was a 92-year-old male with no travel history. The tree topology suggested that the European transmission chain probably started in the UK, and arrived in Italy by means of untraced individuals (Fig. 5 and Supplementary Fig. 4).

Most of SARS-CoV-2 sequences from Lombardy (gray taxa with red dots, 344/346 [99.4%]) are interspersed within B sub-lineages (Fig. 2).

The molecular clock analysis revealed that the most recent ancestor of these viruses dates back to the second half of January (January 20, 95% HPD intervals: Jan, 16–Jan, 23, Supplementary Figs. 4 and 5), in line with evidence of initial outbreaks in other European countries in the second half of January[5]. The Bayesian reconstruction also permitted estimation of the median tMRCA of the COVID-19 pandemic as December 3, 2019 (95% HPD November 19 to December 15, 2019; Fig. 5), consistent with previous estimates[6,7].

By both ML and Bayesian tree, lineages B.1, B.1.1, B.1.8, B.1.1.1, and B.1.5 seem descendant from a sequence belonging to lineage B and collected on January 28 in Germany (known as BavPat1 and named as Jan, 28– Germany in the trees). BavPat1 sequence was part of a B local clade described in Bavaria, resulting from a single travel-associated Chinese primary case[5,8], and singled out as the initial outbreak triggering the Italian one. However, none of the 346 Italian SARS-CoV-2 sequences here analyzed belonged to lineage B, with the exception of one. This sequence differs from BavPat1 for 6 SNP. Thus, according to recent published evidence[9], the lack of a local cluster involving BavPat1 suggested that the initial outbreak in Bavaria was unlikely to have been responsible for directly seeding Lombardy.

Lineage B.1 was initially detected in Lombardy in the second half of February, and in early March was present in the Netherlands, UK, Central Europe, and USA (Supplementary Figs. 4 and 5). Viral sequences were characterized by SNPs at positions 14408 (C to T, non-syn P to L in RdRp) and 23403 (C to T, non-syn D to G in S), with an intra-patient prevalence >95%.

Lineage B.1 was characterized by sustained local transmission, with no evidence of foreign origins, as suggested by the topology of Cluster A. This cluster was characterized by a posterior probability of 0.99 and a tMRCA dated February 14 (Feb, 5–Feb, 20). It was almost exclusively composed of sequences from Lombardy, with the exception of a Swiss 25-year-old patient diagnosed on February 27, a sequence obtained from a Lithuanian patient, whose collection date is partially reported in GISAID (2020-Feb), and a 75-year-old Italian patient diagnosed on March 3 in the Italian Marche region. All Cluster A sequences were characterized by the SNP 26530 (A to G, non-syn D to G in M protein; intra-patient prevalence >90%). The relatively small number of foreign sequences belonging to this cluster, and the strain at the bases of this transmission chain (a sequence collected in Bergamo on March 01), support the assumption, therefore, that the transmission chain started in Lombardy. None of patients involved in this cluster reported international or domestic travels. Two of them (both from Como) stated they had been in contact with a confirmed COVID-19 case, thus confirming the evidence of a local clade.

By taking into account the timing of diagnosis, 40% of the north Italian sequences involved in Cluster A were collected before National lockdown, thus in the early-targeted testing, and 60% in the later-targeted testing (Table 2). The high-risk area of

Bergamo allowed this clade to seed in the time-frame preceding lockdown. On the contrary, Milan, where the perception of infection risk was probably lower and where mobility was permitted until March, was mainly involved at a later time-point, after national lockdown.

Overall, these data suggest that potential origin of lineage B.1 was in Lombardy.

Lineage B.1.1 is a clear descendant of lineage B.1 (posterior probability = 1.00) and was simultaneously detected in late February in Denmark, Germany, UK, United States, and Lombardy (Fig. 5 and Supplementary Fig. 4). Two SNPs in N (28881–28883:GGG to AAC, intra-patient prevalence >99%) further characterized the 126 sequences that belong to this lineage. The exact origin of the entire B.1.1 lineage remains difficult to determine, in the light of its simultaneous circulation in such a vast territory. Two transmission chains probably starting in late February/early March were detected within this lineage (Clusters C and D, Fig. 5, Supplementary Fig. 4, and Table 2). Clusters C and D contained sequences characterized by the non-syn SNP A8072G in NSP3 (N to D), and the syn T4579A in pp1a plus the non-syn G18898T in NSP14 (V to F), respectively. One sequence collected in Como in the first days of March is at the basis of Cluster C, which also included two strains isolated in Belgium thereafter. A sequence diagnosed in Atlanta on February 29 carrying the mutation G18898T but not T4579A is at the origin of Cluster D. Unfortunately, no information regarding recent travel history or contacts are available for these GISAID sequences, neither did patients involved in these clusters report international travel (with the exception of one in Cluster C who had traveled to an unknown destination), making it difficult to reconstruct the initial contact tracing contributing to the definition of this clade.

Overall, the simultaneous circulation of lineage B.1.1 in different countries and the presence of transmission chains with probable foreign origin support the hypothesis that this lineage may have seeded Lombardy by means of diverse geographically distinct access sources.

Thirty-four sequences from Lombardy belonged to lineage B.1.5. This lineage can be divided into two main chains, one of them supported by a posterior probability ≥0.98 (Cluster B). Cluster B involved a complex tracing network of 24 persons, who were infected in Lombardy (mainly in Pavia), and diagnosed after March 8. All sequences were characterized by the SNP 23575 (C to T, non-syn T to I in S protein; intra-patient prevalence: >99%). The earliest and most closely related strain was a sequence from China collected on January 24 in the province of Zhejiang. In line with this, the most recent ancestor of this cluster dates back to January 22 (Jan, 18–Jan, 23), while the transmission chain probably started on February 12 (Feb, 6–Feb, 23) (Fig. 5, Supplementary Fig. 4, and Table 2).

The other chain was composed of a total of 26 sequences (11 from Lombardy), all of them characterized by the SNP at position 20268, A to G, syn (nsp15, intra-patient prevalence: 50.0). Sequences from Lombardy were intermixed with sequences from central and east Europe ($n = 10$), eastern and west Asia ($n = 2$), North America ($n = 1$), UK and the Netherlands ($n = 2$). Looking at the topology of the trees these sequences seem to be closely related to a subgroup of B.1 sequences collected in late February in Lombardy (e.g. Pavia, February 23, and Cremona, February 24 even if the support value is <0.10). The low posterior probability and the absence of information regarding recent travel history makes it hard to understand if a common origin really exists or if their clustering is due to multiple introductions of genetically similar viruses from geographically distinct sources.

Sub-lineages B.1.8 and B.1.1.1 were detected in one and four individuals, respectively. The patient infected by lineage B.1.8 was

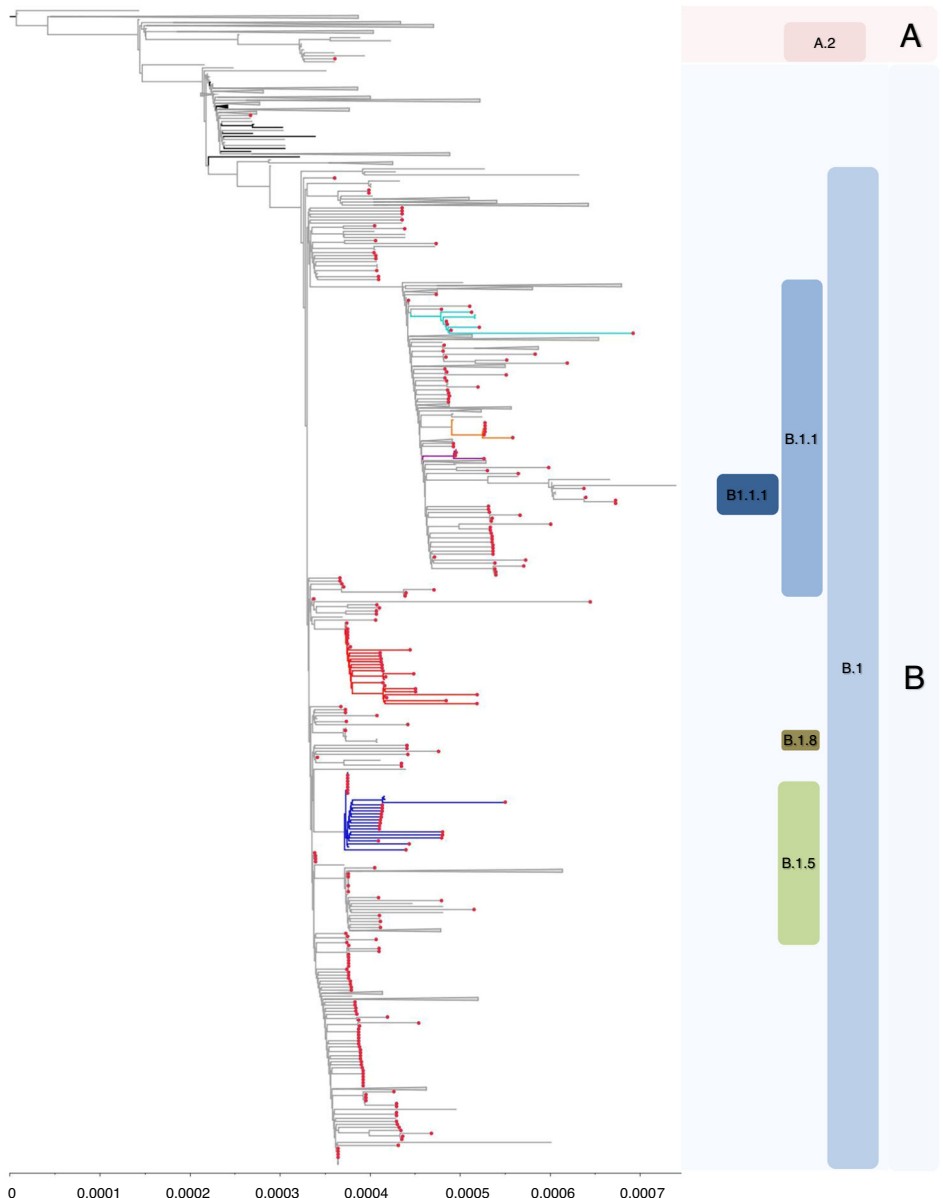

**Fig. 2 Estimated maximum likelihood phylogeny of SARS-CoV-2 genomes from Lombardy (gray taxa with red dots) and genomes from China (black taxa without dots) and other countries (gray taxa without dots), according to lineages.** Local clusters supported by a posterior probability ≥0.98 in the maximum clade credibility tree were highlighted in red (**A**), in blue (**B**), in cyan (**C**), and in orange (**D**). Gisaid sequences come from Italy ($n = 15$), East Europe ($n = 15$), North Europe ($n = 10$), South America ($n = 13$), Africa ($n = 8$), Japan ($n = 10$), Oceania ($n = 25$), West Asia ($n = 17$), South Asia ($n = 14$), Central Europe ($n = 30$), East Asia ($n = 46$), The Netherlands ($n = 41$), South East Asia ($n = 19$), North America ($n = 52$), British Countries ($n = 47$), China ($n = 55$). The phylogeny was estimated with Iqtree with 1000 replicates fast bootstrapping.

an 84-year-old male from Lodi with no travel history, diagnosed on March 5. This infection was preceded by another detected in the Netherlands on March 4 (Fig. 5 and Supplementary Fig. 4, posterior probability = 0.99). This lineage probably originated from B.1 sequences detected in Lombardy in late February (Fig. 5 and Supplementary Fig. 4). Patients infected by lineage B.1.1.1 come from Milan and Pavia, and did not report any travel history. A strain collected in UK on March 13 is at the basis of the B.1.1.1 cluster (Fig. 5 and Supplementary Fig. 4, posterior probability = 1.00), thus suggesting a probable foreign origin for this lineage.

## Discussion

These data on the genomic epidemiology of SARS-CoV-2 in Lombardy, based on a consistent number of whole genome

SARS-CoV-2 sequences circulating in a single region, indicate the simultaneous circulation of at least three widely represented lineages of SARS-CoV-2 (B.1, B.1.1, and B.1.5), supported by transmission chains occurring since the first half of February (Table 2, Fig. 5, Supplementary Fig. 4).

Even if these lineages were sizeable and spread rapidly, they showed different geographical distributions. Lineage B.1, probably characterized by origin in Lombardy, mostly affected Lodi, Cremona, and Mantua in the south, and accounted for 90.2% of sequences coming from these provinces (55/61, Fig. 1 and Table 1). This is a less densely populated area, with less intense interconnections with Europe and beyond, making a modest lineage diversification plausible. Lineage B.1.1 predominated in the north of Lombardy, mostly in Como, Lecco, Bergamo, and its adjacent territories such as Alzano and Nembro, and accounted

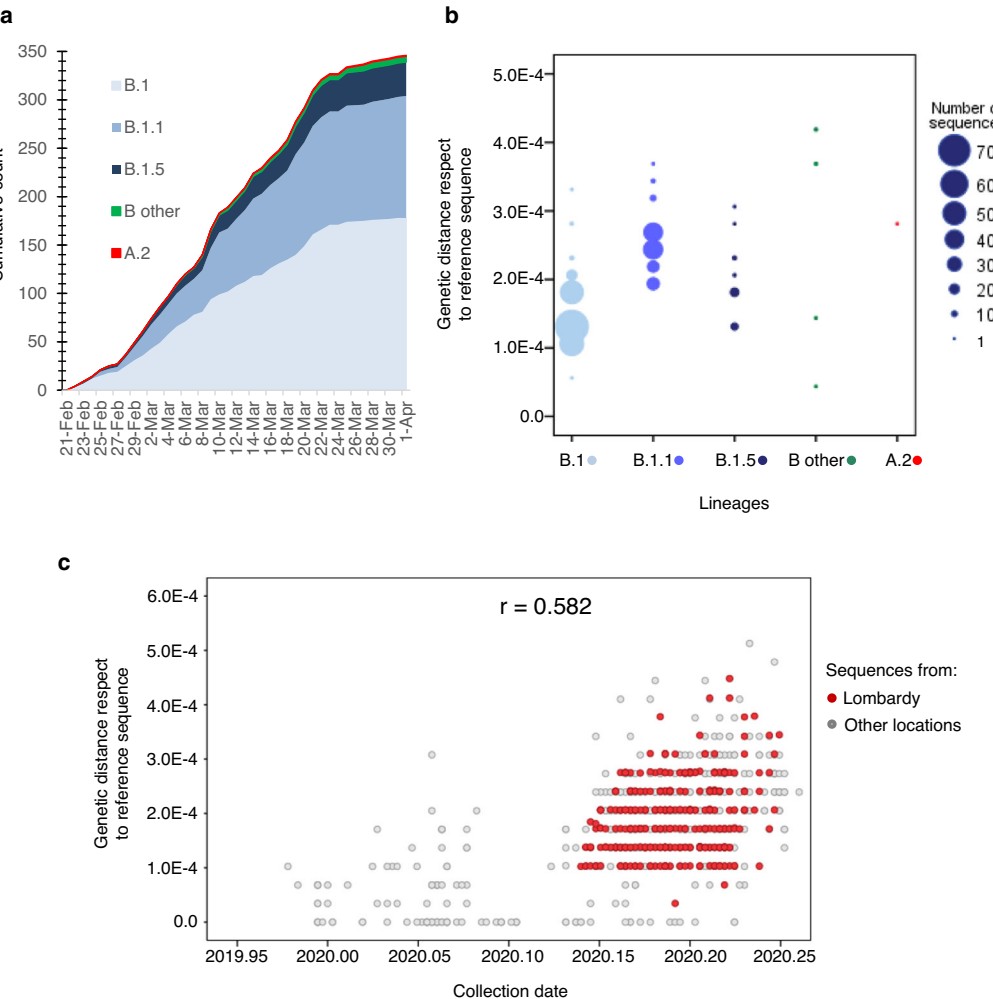

**Fig. 3 Distribution of SARS-CoV-2 genomes against lineages and collection date. a** Cumulative counts of the 346 SARS-Cov-2 genomes circulating in Lombardy against collection date. **b** Genetic distance for the Sars-Cov-2 genomes circulating in Lombardy against lineages. **c** Genetic distance for the Sars-Cov-2 genomes against collection date. The Pearson correlation coefficient between genetic distance and collection date is displayed in the top-right corner ($r = 0.582$). Sequences are colored by sampling location (Lombardy = red, other location = gray). Reference sequence: NC_045512.2. 'B other' refers to haplotypes B, B.1.1.1, and B.1.8.

for 62.7% of sequences coming from these provinces (84/134, Fig. 1 and Table 1). This is a densely populated area, served by dense motorway and railway networks, and by two international airports. Thus, we cannot exclude that foreign introductions contributed to the spread of this lineage, a hypothesis supported by the characteristics of the local clusters identified within it. Lastly, lineage B.1.5 was mainly detected in Pavia, because of a local cluster detected in this area which involved 23 B.1.5 sequences. Of note, Milan, the most important urban area of Lombardy, is the only province in which all lineages are equally represented (Table 1, $P = 0.108$ by Chi-square test for trend). Given the central location of Milan and its intense economical and transportation links to Europe and beyond, the simultaneous circulation of the different lineages in this area would seem plausible.

These data are in line with recent accounts of the dynamics of the SARS-CoV-2 pandemic in densely populated areas like New York State, Belgium, and (more recently) the Netherlands[10–12]. The same scenario was also observed in less densely populated areas characterized by regular international travel, such as Iceland[13], where the government managed to contain the spread of

the virus thanks to a strategy of aggressive testing, contact tracing, and quarantine. To contain and mitigate the COVID-19 epidemic, Italy has been the first European country to implement unprecedented measures to restrict individual mobility, and to promote social distancing, with the aim of interrupting transmission of the SARS-CoV-2 virus. Following the detection of the first case of COVID-19 cases in Lombardy on February 21, the government implemented an ever-increasing number of emergency decrees, which culminated in the complete lockdown on March 8.

Due to the intense commercial relations between China and the rest of the world, thousands of Chinese and Europeans traveled from China to Europe and vice versa during January. In Italy, direct flights from China were stopped on January 31, while indirect access from China through other routes was suspended only later (see below). However, SARS-CoV-2-infected people could enter Italy long before this date, as was proven by the diagnosis of SARS-CoV-2 in Chinese tourists who arrived in Italy on January 23. In addition, the suspension of direct flights from China could not prevent the entrance of people though transit flights from other countries[14]. The analysis of the COVID-19

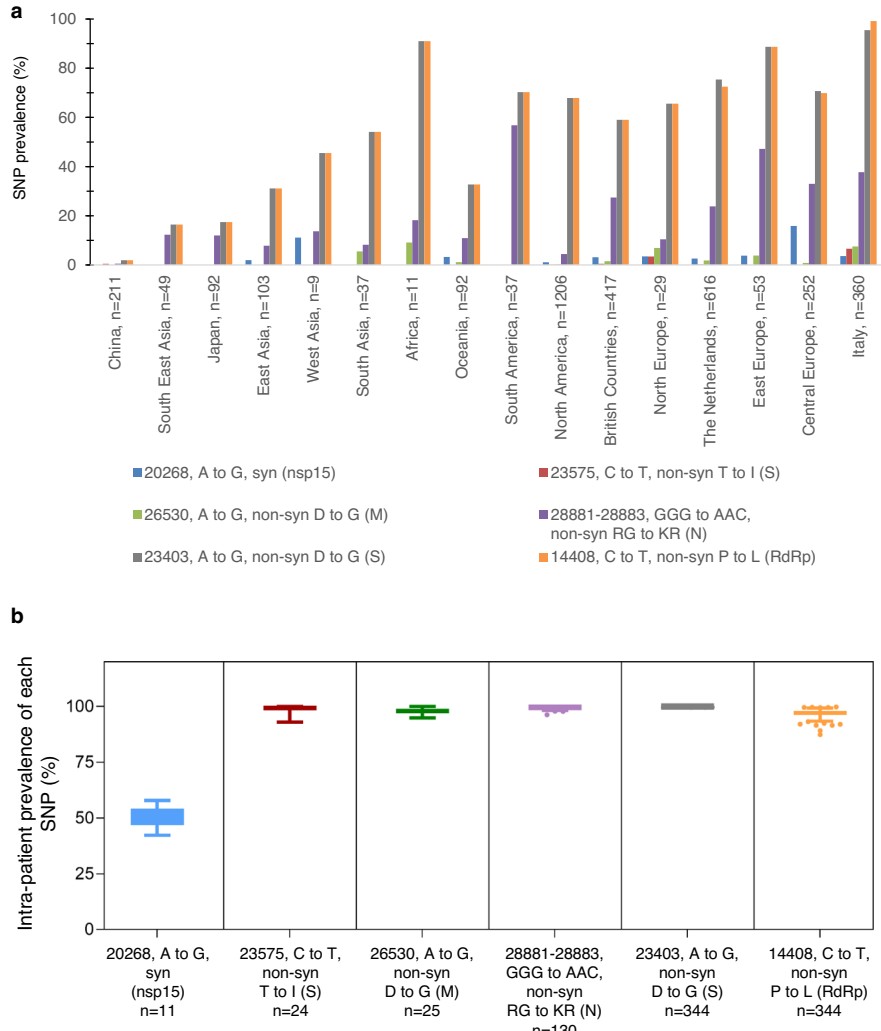

**Fig. 4 Prevalence of most representative single nucleotide polymorphisms (SNPs) in SARS-CoV-2 genomes isolated in Lombardy according to geographical locations and their intra-patient prevalence. a** Frequency of single nucleotide polymorphisms (with respect to the Wuhan reference genome NC_045512.2) among SARS-CoV-2 sequences according to different geographical location. Italian sequences include the 6 and 8 sequences from North and Central Italy present in Gisaid at April 10. **b** Intra-patient prevalence of most representative SNPs in North Italian SARS-CoV-2 sequences according to sample date. Box and whiskers plots indicate the median and interquartile range of intrapatient prevalence. SNP: single nucleotide polymorphism; syn: synonymous; non-syn: non-synonymous; nsp15: non-structural protein 15; S: spike; M: matrix; N: nucleocapsid; RdRp: RNA-dependent RNA polymerase.

outbreak and the modeling assessment of the effects of travel limitations also confirmed that, despite the comprehensive restrictions on travel to and from China since January 23, 2020, many individuals exposed to SARS-CoV-2 have been traveling internationally without being detected[15]. Lombardy, with three international airports (e.g. Milano, Linate, Bergamo), is Italy's greatest and most prosperous international business hub, constantly hosting international events. It is thus very likely that a number of infected people, both directly from China and also through other countries, entered Lombardy in that period[16]. The untraced entry of SARS-CoV-2 might have facilitated the diffusion of the virus at population level and its spread at local level, as suggested by the epidemiological characteristics of the identified local clusters, mainly sustained by patients with no travel history.

Our data allow accurate interpretation of the COVID-19 outbreak in Lombardy and its transmission dynamics also, thanks to a revised phylogenetic approach, although some national-level data have been published (with limited sampling and details)[17]. In particular, the circulation of SARS-CoV-2 in the country

(an area of 300,000 km$^2$, and a population of 60 million) was described by analyzing a lower number of whole-genome sequences. Here, we focused the attention on a territory of 24,000 km$^2$, with a population of 10 million, which accounted for 37% of COVID-19 cases and 53% of deaths in the country. Lombardy is also the first region to have been affected by the COVID-19 outbreak, with a rate of 112.9 deaths per 100,000 population, almost six times higher than in the rest of Italy. The first phase of epidemic also saw the emergence of many cases within an extremely short period of time, a scenario that does not necessarily favor the simultaneous and massive circulation of different lineages. The effort to sequence and implement with epidemiological data 346 SARS-CoV-2 sequences endemic in this territory made it possible to describe the heterogeneity of the virus circulating in Lombardy, and ultimately to appreciate local sustained transmissions and to estimate the time of their probable introduction.

Looking at the correlation between lineages and COVID-19, no trend of association between disease severity (including evidence

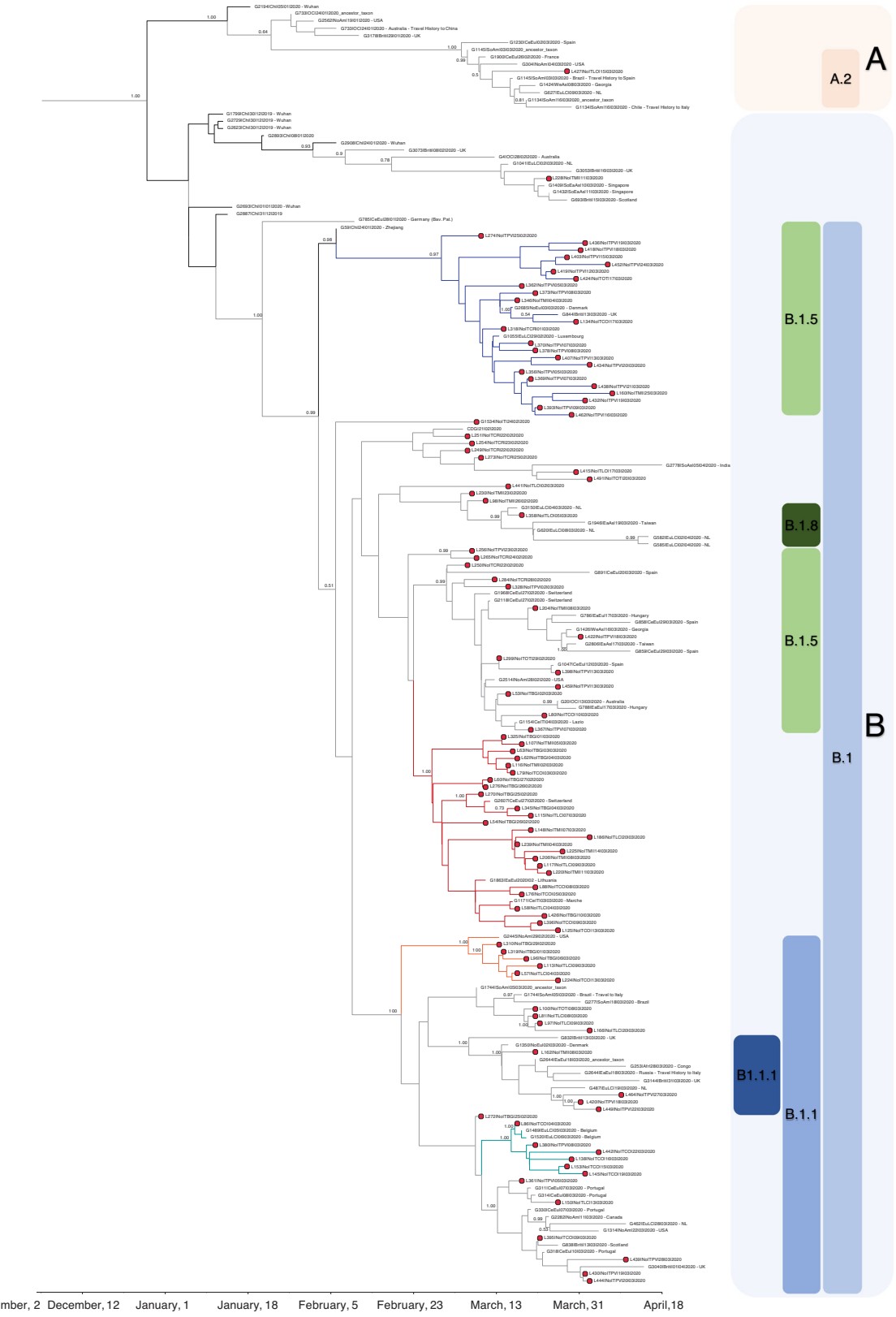

**Fig. 5 Bayesian phylogeographic reconstruction incorporating location and travel history data of the most informative sequences for virus spread and clustering in Lombardy.** SARS-CoV-2 genomes from Lombardy were shown by red circles. Local clusters of SARS-CoV-2 sequences supported by a posterior probability ≥0.98 by the time-scaled maximum clade credibility tree were highlighted in red (**A**), in blue (**B**), in cyan (**C**), and in orange (**D**). Two independent chains were run for 25 million states. Parameters and trees were sampled every 1000 states.

**Table 2 Distribution of SARS-CoV-2 local clusters, according to lineages, timing of diagnosis, and reported risk.**

| | tMRCA[a] | Earliest and most closely related strains outside Lombardy[b] | Before March 8 | | | | | After March 8 | | | | |
|---|---|---|---|---|---|---|---|---|---|---|---|---|
| | | | N (%) | Age, median IQR | Provinces, N (%) | International Travels, N (%)[c] | Covid-19 contact, N (%)[d] | N (%) | Age, median IQR | Provinces, N (%) | International Travels, N (%)[c] | Covid-19 contact, N (%)[d] |
| **Lineage B.1** Cluster A | Feb, 14 (Feb, 5–Feb, 20) | None | 10 (40.0) | 66 (47–78) | Bergamo: 8 (80.0) Como: 1 (10.0) Lecco: 1 (10.0) Milan: 0 (0.0) | 0 | 0 | 15 (60.0) | 58 (51–80) | Bergamo: 1 (6.6) Como: 4 (26.7) Lecco: 3 (20.0) Milan: 7 (46.7) | 0 | 2 |
| **Lineage B.1.5** Cluster B | Feb, 12 (Feb, 6–Feb, 23) | China: 1 (Zhejiang) | 9 (39.1) | 70 (61–86) | Cremona: 1 (11.1) Milan: 1 (11.1) Pavia: 7 (77.8) | 0 | 0 | 14 (60.9) | 78 (60–86) | Como: 1 (7.1) Milan: 1 (7.1) Pavia: 11 (78.6) Other: 1 (7.1) | 0 | 3 |
| **Lineage B.1.1** Cluster C | March, 1 (Feb, 27–March, 6) | None | 2 (33.3) | 36 | Pavia: 1 (50.0) Como: 1 (50.0) | 0 | 0 | 4 (66.7) | 81 (64–81) | Como: 4 (100.0) | 0 | 0 |
| Cluster D | Feb, 27 (Feb, 20–Feb, 28) | NoAm: 1 (Atlanta) | 3 (50.0) | 80 (51–82) | Bergamo: 1 (33.3) Como: 1 (33.3) Lecco: 1 (33.3) | 0 | 0 | 3 (50.0) | 68 (37–79) | Bergamo: 2 (66.7) Lecco: 1 (33.3) | 0 | 1 |

Time of the most recent common ancestors and most closely related strains were also reported for each cluster.
[a]Time of the most recent common ancestors (TMRCA) of lineages and local clusters characterized by a posterior probability ≥0.98 in the Bayesian Tree. TMRCA are represented in the order of date/month/year. The values in parentheses represent the 95% HPD intervals.
[b]The SARS-CoV-2 genomes most closely related to lineages and clusters identified in Lombardy by the Bayesian Tree.
[c]Data available for 51 patients.
[d]Data available for 24 patients.

of interstitial pneumonia, and severe or critical presentation) and lineages was detected (Table 1).

Looking at the overall variability of SARS-CoV-2, we found that only six SNPs (five out of six non-synonymous)[18–20] characterized our consensus sequences, highlighting a good conservation rate of the virus over time. This conservation rate is confirmed within the spike structural protein, where only two mutations, one of them at low prevalence (i.e, C23575T, corresponding to the amino acid variant T671I), were detected. None of these mutations have a role in altering pre-existing N-glycosylation sites or in creating new ones[21], a factor which may be beneficial for the development of vaccines strategies.

Worthy of mention is the SNP A to G at position 23403, corresponding to the variant D614G in spike protein, detected in 99.4% of our SARS-CoV-2 sequences (all with an intra-patient prevalence >99%). As is already known, this variant is observed frequently in European countries, such as the Netherlands, Switzerland, and France, but seldom observed in China. As recently reported, this variant, located within a B-epitope, causes the substitution of a large acidic residue (aspartic acid), with a small hydrophobic residue (glycine). Its rapid fixation at population level might thus suggest a role in viral entry, and enhancement of interaction between receptor-binding-domain of the S protein with the entry receptor ACE2 (ref. [22]). In line with this evidence, D614G variant was recently associated with lower Ct values, indicative of potentially higher in vivo viral loads[23]. The presence of D614G in almost all the sequences sampled in Lombardy suggests the dominance of this mutation since the first phases of epidemic in Lombardy, and in Europe, confirming its rapid spread and persistence. Unfortunately, the fixation of this mutation in our region prevented us from defining any association with disease status and Ct values. However, the quite low Ct values observed in our population (median, IQR: 18.8, 16.8–20.1) could be in line with the association of D614G variant with lower Ct values, indicative of potentially higher in vivo viral loads[23], and thus related to severe outcome in COVID-19 (refs. [24–26]).

Our study has some limitations. The analysis of phylogenetic structures during such an early phase of the pandemic should be interpreted carefully, as the number of mutations that define phylogenetic lineages is small and may be similar to the rate of potential errors introduced during reverse transcription, PCR amplification, or sequencing[27]. To overcome these problems, Bayesian approach, recognized as a powerful way to estimate species divergence[28], and thus expected to provide more robust results, was applied. Moreover, the integration of host characteristics (such as geographical location, collection date, and clinical manifestations) aided phylogenetic interpretation.

The intra-host variability of SARS-CoV-2, and the role of potential existing minority variants, has not been investigated here. Initial evidence suggests that intra-host variation of SARS-CoV-2 can be frequently found among clinical samples (median number of intra-host variants: 1–4), but at the same time these variants were not observed in the population as polymorphisms, probably suggesting a bottleneck or purifying selection was involved[29,30]. Thus, ad hoc designed studies are necessary to provide an extensive overview of SARS-CoV-2 intra-host variability and minority variants description, if and how these minority variants can spread in the population, and their potential role in virulence and transmissibility.

In the peak of SARS-CoV-2 epidemic, diagnosis was mainly addressed to symptomatic cases (prevalently older people with no travel history) or subjects at high risk of exposure (i.e. health care workers exposed to positive patients without adequate protection). This approach may have caused a substantial underestimation of positive subjects, precluding the possibility of

extending our analysis to asymptomatic infections, whose role in the transmission chains and in influencing the evolution of epidemic remains a challenge to investigate.

Moreover, even if the number of samples here analyzed is noteworthy, the epidemic in the east (i.e. Brescia and Mantua) and the valleys of the north (i.e. Valtellina and Valcamonica, Fig. 1) is poorly represented. Yet, with the sole exception of Brescia, which resembled Bergamo, Cremona, and Lodi in how SARS-CoV-2 spread, Mantua and the valleys accounted for only the 6.7% of confirmed COVID-19 cases in Lombardy, a percentage that does not constitute a major issue for the good representability of SARS-CoV-2 epidemic in the region. Even if the sampling scheme used in this study (Supplementary Fig. 1, Supplementary Table 1, and Supplementary Results) suggests a good representativeness of the selected population in relation to the general population, we cannot exclude that denser sampling (both temporally and spatially) would reveal novel dispersal patterns not observed here and further address lineages movement.

In conclusion, this study contributes to the identification and circulation pattern of seven SARS-CoV-2 lineages in an area highly affected by COVID-19, which caused more than 16,000 deaths in Lombardy in a number of weeks. Of these seven lineages, three were widely represented and at least two likely originated in this region, suggesting that the virus was circulating undetected for some time before first detection and confirming a central role of Lombardy in the SARS-CoV-2 epidemic. We cannot exclude that this multiple and simultaneous circulation of SARS-CoV-2 strains can have exacerbated the transmissibility potential of the virus and thus created a real viral storm in such a densely populated region.

## Methods

**Sample collection and epidemiological data**. This retrospective observational study included 371 SARS-CoV-2-positive nasopharyngeal-swabs obtained from adult patients hospitalized or referred for the diagnosis at two major hospitals in Lombardy from February 22 to April 4. Demographics, epidemiological and clinical data were obtained retrospectively by pseudonymized electronic medical records. The study protocol was approved by local Research Ethics Committee of the Niguarda and San Matteo hospitals (prot. 92-15032020 and P_20200029440). This study was conducted in accordance with the principles of the 1964 Declaration of Helsinki. Informed consent was waived in accordance with Italian governmental regulations on observational retrospective studies[31].

The severity of the disease was classified in line with WHO scale[32] as: (i) mild: symptomatic patients meeting the case definition for COVID-19 without evidence of viral pneumonia or hypoxia; (ii) moderate: clinical signs of pneumonia (fever, cough, dyspnea, fast breathing) but no signs of severe pneumonia, including $SpO_2 \geq 90\%$ on room air; (iii) severe: clinical signs of pneumonia (fever, cough, dyspnea, fast breathing) plus one of the following: respiratory rate >30 breaths/min; severe respiratory distress; or $SpO_2 < 90\%$ on room air; (iv) critical: presence of Acute Respiratory Distress Syndrome (ARDS), and/or sepsis or multiorgan failure (septic shock).

**Virus amplification and sequencing**. Total RNAs were extracted from nasopharyngeal swabs by using QIAamp Viral RNA Mini Kit, followed by purification with Agencourt RNAClean XP beads. Both the concentration and the quality of all isolated RNA samples were measured and checked with the Nanodrop. Virus genomes were generated by using a multiplex approach, using version 1 of the CleanPlex SARS-CoV-2 Research and Surveillance Panel[33,34], according to the manufacturer's protocol starting with 50 ng of total RNA and followed by Illumina sequencing on a NextSeq 500. Briefly, the multiplex PCR was performed with two pooled primer mixtures and the cDNA reverse transcribed with random primers was used as a template. After ten rounds of amplification, the two PCR products were pooled and purified. Then the digestion reaction was performed to remove non-specific PCR products, followed by second PCR reaction for barcoding with 24 rounds of amplification. Libraries were checked using High Sensitivity Labchip and quantified with Qubit Fluorometric Quantitation system (Life Technologies). Equimolar quantity of libraries was pooled, and the obtained run library mix was loaded at 1.5 pM into NextSeq500 for sequencing in the Mid Output format with paired-end $2 \times 150$ bp. The Illumina sequencing platform takes <26 h to obtain

30.2 Gb of sequencing data (compressed format), achieving between 170.000 and 856.000 paired-end fragments per sample (340.000 and 1.712.000 sequences), with a mean coverage depth of 2.500.

**Virus genome assembly**. Reference-based assembly of the raw data was performed as follows: Illumina adapters were removed, and reads were filtered for quality (average q28 threshold and read length >135 nt) using FASTP[35]. First and last 15 nucleotides were then removed from all reads. The mapping of cleaned reads was performed against the GenBank reference genome NC_045512.2 (Wuhan, collection date: December 2019) using BWA-mem[36]. Reads mapping on SARS-CoV-2 reference genome were a median of 99.7% (IQR: 99.5%–99.8%), and were able to cover from 94.0% to 99.7% of the SARS-CoV-2 reference genome (GenBank: NC_045512.2), independently of SARS-CoV-2 load (Supplementary Fig. 2a, b). The few genome regions ($N = 4$) with lower reads coverage were consistently limited to no more than 35 nt. SNP variants were called through a pipeline based on samtools/bcftools[37], and all SNPs having a minimum supporting read frequency of 40% with a depth $\geq 50$ were retained. Consensus was generated using the github freely distributed software vcf_consensus_builder (https://github.com/peterk87/vcf_consensus_builder).

**Phylogenetic analysis**. Sars-CoV-2 lineages of the 346 SARS-CoV-2 consensus sequences obtained were assigned according to the PANGOLIN application (Pangolin, https://pangolin.cog-uk.io/)[4]. In order to represent the global diversity of the lineages by the end of April 2020 while minimizing the impact of sampling bias, 395 GISAID deposited sequences were added to the consensus sequences obtained by our samples. Sequences were aligned using ClustalX and manually inspected in Bioedit. The final alignment was composed of 741 sequences 29,159 nucleotides long.

To account for regions which might potentially be the result of hypervariability or sequencing artifacts, alignment positions showing significant homoplasy were identified by a combined approach. Homoplasies were firstly identified using HomoplasyFinder, and then confirmed by Treetime (homoplasy setting)[38,39].

In order to explore the phylogenetic structure of SARS-CoV-2, the 741 sequences were analyzed by the maximum likelihood (ML)[40–42] and Bayesian coalescent[43,44] methods, under the best-fit model of nucleotide substitution GTR+I[41]. The Bayesian coalescent tree analysis was undertaken with BEAST v1.10.5 (ref. [43]), using the GTR+I substitution model with an exponential population growth tree prior and strict molecular clock, under a noninformative continuous-time Markov chain (CTMC) reference prior[44]. The information regarding location and recent travel history (less than 1 month before the SARS-CoV-2 diagnosis) of the most informative sequences for virus spread and clustering identified in the first Bayesian tree were incorporated in a second Bayesian tree interference[45], in order to yield more robust reconstructions of virus spread.

For further details regarding the homoplasy checking, the criteria used for the selection of GISAID sequences, and phylogenetic methods, refer to Supplementary Methods.

**Statistical analysis**. Data were analyzed using Rgui and the statistical software package SPSS (v32.0; SPSS Inc., Chicago, IL).

**Reporting summary**. Further information on research design is available in the Nature Research Reporting Summary linked to this article.

## Data availability

Nature Research reporting summary, source data, and supplementary information were uploaded as related manuscript files. The 346 SARS-CoV-2 sequences obtained in this study are openly available on GISAID portal and European Nucleotide Archive under the accession numbers EPI_ISL_542098-EPI_ISL_542443 and ERX4545164-ERX4548732, respectively. The list of accession numbers is available in the Supplementary Data 3. The list of the whole-genome SARS-CoV-2 sequences ($n = 3244$) retrieved from GISAID (gisaid.org) on 3 May 2020 and their corresponding accession numbers and lineages are available in the Supplementary Data 1. The list of the 395 selected GISAID SARS-CoV-2 sequences considered for phylogenetic analyses and their corresponding accession numbers and lineages are available in Supplementary Data 2. The list of the significant homoplastic positions identified by HomoplasyFinder and TreeTime are available in the Supplementary Data 4. Lombardy map was retrieved from https://raw.githubusercontent.com/blackmad/neighborhoods/master/lombardy.geojson. COVID-19 cumulative case data in Lombardy were retrieved from the Italian Ministry of Health Services COVID-19 dashboard (http://www.salute.gov.it/portale/nuovocoronavirus/dettaglioContenutiNuovoCoronavirus.jsp?lingua=english&id=5367&area=nuovoCoronavirus&menu=vuoto). General population data were retrieved from the Italian National Institute of Statistics (http://dati.istat.it/Index.aspx?DataSetCode=DCIS_POPRES1). The de-identified data regarding demographic and clinical features related to each patient are available on reasonable request from the corresponding author. Data are not publicly available in order to fully comply to privacy guarantee. Source data are provided with this paper.

## Code availability

Beast.xml files used to infer the corresponding time-scaled maximum clade credibility tree and the Bayesian tree interference are available at https://doi.org/10.5281/zenodo.4306802.

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

## Acknowledgements

We thank Cariplo Foundation that financially supported this work by an unrestricted grant. We thank Biodiversa s.r.l. for providing technical support, particularly Dr. Marco Dotto for his valuable assistance in the implementation of the study. We thank F. McDonald for English discourse revision and editing. The authors also thank Dr. Silvia Nerini and the whole staff of the Microbiology and Virology Laboratory of ASST Grande Ospedale Metropolitano Niguarda and IRCCS San Matteo for outstanding technical support in processing swab samples, performing laboratory analyses, and data management.

## Author contributions

C.A. conceived the study design and data analysis and wrote the manuscript; V.Ce. and A.P. helped in data collection, data analysis, data interpretation, and writing; V.Co. performed the bioinformatic analysis and data processing; M.T., L.C., S.C., S.R., F.G., and F.N. processed the samples and collected the data; S.G. helped in bioinformatic analysis; E.M. and M.A. helped in the sample processing and data collection; C.V., R.F., O.M.E., and M.P. recruited the samples and enrolled patients; C.F.P. and F.B. conceived and directed the study, and critically revised the manuscript.

## Competing interests

The authors declare no competing interests.
