## [Peer Review File · Nature Communications]

REVIEWER COMMENTS

Reviewer #1 (Remarks to the Author):

The authors report a large genomic analysis of SARS-CoV-2 in Lombardy, Italy, and likely represents a considerable effort by the authors. The dataset is accompanied by a clinical evaluation of severity, categorised as mild, moderate or severe.

The conclusion that there were multiple introductions of the virus to Lombardy is justified by the phylogenetic analysis - although the source of those introductions is unclear as the paper stands. This is in keeping with similar reports in other European countries, the USA and more recently in Africa. Further, it seems likely that the molecular clock analysis reflects earlier introduction(s) into Lombardy.

I think the paper would be strengthened by a number of modifications and additional analyses.

1. While multiple introductions seems to be a theme of the spread of this virus, the origins of those introductions remain unclear and are of great importance in understanding the rapid movement of the virus across Europe. The authors have not displayed the likely origin of the lineages that they report, either from accompanying epidemiological data (despite a retrospective review of the clinical notes) or from the phylogenetic analysis. If these data could be incorporated, particularly travel history, the paper would be far more informative and original in nature. Some commentary on travel in the region - flights in particular would also be useful.

2. With reference to the above comment, the phylogenetic analysis would benefit from alignment with the proposed global lineage derivation system Pangolin <https://pangolin.cog-uk.io/> (or a suitable equivalent) so that the lineages found can be compared with others on the global tree. A more minor point, it is not discussed how homoplasy is dealt with in the analysis.

3. The D614G mutation is mentioned in the text, but the authors have not carried out an analysis of how this relates to clinical outcome or CT - this would be a useful analysis since they have gone to the trouble to categorise severity of the disease by lineage (I'm not sure that the clinical tool used is the best choice - the ordinal WHO scale might be a better one).

4. It was not clear how the samples analysed were selected (was this randomised selection of the total, or governed by CT value or availability?), nor how they relate to the incidence of infection in different parts of Lombardy. Figure 1 shows a nice geographical coverage but would be strengthened by a second panel showing the numbers diagnosed in each area.

I would have been interested to see the sequence data - the authors plan to upload to GISAID - would be useful if this could be uploaded (useful also to the community in general).

Reviewer #2 (Remarks to the Author):

This study on the genomic epidemiology of SARS-CoV-2 in Italy attempts to uncover the introduction and transmission of the virus in Lombardy from February to April 2020. Lombardy was one of the hardest hit regions within Italy and was a potential hotspot for transmission to the rest of Europe. At a first glance this study is potentially interesting as it examines some of the earliest SARS-CoV-2 infections from Italy and thereby from a phylogenetic perspective it could fill some missing gaps in the early course of the pandemic within Europe. However, there are many issues

that detract from the potential of this study and some of the conclusions are not fully justified. One of my major concerns is the fact that it is not clear that these sequences actually currently exist on GISAID. In fact, a quick look at GISAID reveals that this sequence data is not currently uploaded. While this is the author prerogative, I find it is somewhat troubling that the authors have not shared this data given its potential importance during a pandemic crisis. This aside the idea that there are multiple introductions derived from two major lineages is not overly novel and I wonder why the authors have not expanded on this to suggest where the introductions were derived from. Most studies to date have shown that the virus was introduced on multiple independent occasions so why would Italy be any different. Also, in the absence of sequence data from China added to the phylogenetic trees I would suggest that stating two introductions is being very conservative as it is more likely that there have been many more independent introductions from China and mainland Europe.

Major comments:

1. Greater clarity and investigation should be performed on the role that Italy played in the early establishment of SARS-CoV-2 in Europe as there is still debate on whether an initial outbreak in Bavaria, Germany seeded the Italian outbreak. (See <https://www.biorxiv.org/content/10.1101/2020.05.21.109322v1> for a full discussion). I find it odd that the authors had not discussed this in any great detail.
2. The authors findings that SARS-CoV-2 was present in the region one month before the first diagnosis in Lodi is not proof that it was circulating in the region as it merely reflects the time that the ancestor was circulating somewhere across the globe. The TMRCA is the date of the common ancestor of the sampled genomes in a transmission lineage. While the TMRCA represents the earliest transmission event in the lineage revealed by the data, it does not necessarily represent the first transmission event in the lineage as a whole. Specifically, if the transmission lineage is well sampled then the TMRCA represents the date of the first transmission event in the UK lineage. However, if the transmission lineage is poorly sampled then the TMRCA may represent a later transmission event in the lineage. The actual tMRCA for only Italian sequences is in fact much later for clusters A and B according to Figure 4 See Figure 1 from <https://virological.org/t/preliminary-analysis-of-sars-cov-2-importation-establishment-of-uk-transmission-lineages/507>
3. The authors state that the sequence data from Figure 3 and 4 does not include those sequences derived from China as they wanted to examine transmission chains. Removing sequence data from China is not the best way to do this as they may be artificially creating transmission chains. If the authors added sequence data from Wuhan and elsewhere in China how do these results change?
4. Why did the authors choose metagenomic sampling? What percentage of reads were SARS-CoV-2? Were there other co-infections? If the authors use such approach, they should go into some level of detail on their sequencing findings such as the above.
5. Lineages and sequences should be genotyped according to the PANGOLIN application by Rambaut et al. (<https://www.biorxiv.org/content/10.1101/2020.04.17.046086v1>) as this is becoming the universal classification scheme and it will help orientate the readers.
6. Sampling bias and variation. What sampling criteria and justification was used for selecting 371 samples from 11,445 COVID-19 positive specimens? This only represents 3% of the positive population so I am concerned why the authors did not choose more samples and are this small subset representative of the age distribution? Table 1 could be expanded to look at the characteristics of these 11,445 and statistics used to evaluate whether there was any significant difference between those sequenced and those not sequenced.
7. Similarly, as stated above any statistical differences in the p-values between lineages 1 and 2 could simply reflect a sampling choice and not truly reflect the dynamics of the virus.
8. Is there any epi data to support these clusters A and B. Is it not conceivable that infected individuals for instance in cluster B were all infected elsewhere (outside Italy) from a common source and then these cases were imported giving the false impression of an Italian cluster.
9. Why did the authors use the "best-fit model of nucleotide substitution GTR+I8 with gamma-distributed rate" for ML tree reconstruction but HKY+Q4 was used in BEAST? Why was not the same model implemented in BEAST. This is confusing and should be clarified. XML's file (minus the sequences should be provided).

10. How are the authors sure that the measured diversity illustrated in the supplementary figures is not measuring re-introduction of the virus from travel cases?
11. The manuscript is devoid of some level of detail on their global dataset. GISAID numbers should be given as a supplementary table to illustrate the breakdown of sequences per country. As the phylogenetic results may be somewhat sensitive to how these sequences were selected.
12. Where any sites in the alignment masked for issues related to homoplastic substitutions (see <https://virological.org/t/issues-with-sars-cov-2-sequencing-data/473>).
13. Based on sequence data it is difficult to equate genotype to phenotype and the suggestion that because lineages have similar evolutionary characteristics then they have similar replication profiles is unfounded and should be removed (line 172).
14. Have the authors examined D614G spike mutation that has been speculated to be linked with disease severity.

Response to Reviewers' Comments

Reviewer #1

Remarks to the Author:

The authors report a large genomic analysis of SARS-CoV-2 in Lombardy, Italy, and likely represents a considerable effort by the authors. The dataset is accompanied by a clinical evaluation of severity, categorised as mild, moderate or severe. The conclusion that there were multiple introductions of the virus to Lombardy is justified by the phylogenetic analysis - although the source of those introductions is unclear as the paper stands. This is in keeping with similar reports in other European countries, the USA and more recently in Africa. Further, it seems likely that the molecular clock analysis reflects earlier introduction(s) into Lombardy.

I think the paper would be strengthened by a number of modifications and additional analyses.

1. While multiple introductions seem to be a theme of the spread of this virus, the origins of those introductions remain unclear and are of great importance in understanding the rapid movement of the virus across Europe. The authors have not displayed the likely origin of the lineages that they report, either from accompanying epidemiological data (despite a retrospective review of the clinical notes) or from the phylogenetic analysis. If these data could be incorporated, particularly travel history, the paper would be far more informative and original in nature. Some commentary on travel in the region - flights in particular would also be useful.

Answer: To explore the distribution of SARS-CoV-2 sequences in Lombardy and to shed further light on the geographic origin of the SARS-CoV-2 infections in residents of Lombardy, we performed both an estimated maximum likelihood (ML) phylogeny, and a Bayesian molecular clock analysis of 741 viral sequences (346 from complete viral genomes from Lombardy population and 395 from GISAID). To improve the epidemiological characterization of virus spread and local clusters origin, the available information regarding recent travel history or contacts were retrieved from sequences collected in Lombardy (available information for 237 individuals) and GISAID. The information regarding location and recent travel history of the most informative sequences for virus spread and clustering identified in the first Bayesian tree were incorporated in a Bayesian tree inference (Lemey P et al., bioRxiv 2020), in order to yield more realistic reconstructions of virus spread.

According with these revised and additional analyses, the revised version of the manuscript now includes a revised version of Figure 4 (a Maximum Likelihood tree where the lineages distribution is shown), a new Supplementary Figure 4 (a Bayesian Tree where the lineages distribution, their most likely origin, and clusters supported by a posterior ≥ 0.98 are shown), a new Figure 5 (a Bayesian Tree incorporating information regarding location and recent travel history) and a new Table 2, reporting the distribution of SARS-CoV-2 local clusters, according to lineages, time of the most recent common ancestors, most closely related strains, timing of diagnosis, age, location and reported risk.

By reassigning sequences to the lineages according to the PANGOLIN application, we found that the majority of sequences ($n=179$, 51.7%) belonged to lineage B.1, 126 (36.4%) belonged to lineage B.1.1, and 34 (9.8%) were assigned to lineage B.1.5. Lineage B.1.1.1 was found in four North Italian sequences, while A.2, B and B.1.8 lineages were detected in one patient each.

The patient infected by lineage A.2 was diagnosed on March 15 in Lodi. Unfortunately, nor travel information nor possible COVID-19 contact were available for this patient, thus hampering the tracing of the exact dynamic of this infection. Looking at the topology of the trees, the European transmission chain probably originated in Spain (Figure 5 and Supplementary Figure 4) and

continued in France in the second half of February. The patient infected by lineage B was a 92 years old male with no travel history. The tree topology suggested that the European transmission chain probably started in UK, and arrived in Italy by untraced individuals (Figure 5 and Supplementary Figure 4).

Most of SARS-CoV-2 sequences from Lombardy (grey taxa with red dots, 344/346 [99.4%]) are interspersed within B sub-lineages (Figure 4).

The molecular clock analysis revealed that the most recent ancestor of these viruses dates back to the second half of January (Jan, 20, 95% HPD intervals: Jan, 16 – Jan, 23, Supplementary Figure 4 and Figure 5), in line with evidences of initial outbreaks in other European countries, in the second half of January (Böhmer MM et al., 2020).

By both ML and Bayesian tree, lineages B.1, B.1.1, B.1.8, B.1.1.1 and B.1.5 seem descendant from a sequence belonged to lineage B and collected on January 28 in Germany (known as BavPat1 and named as Jan, 28 – Germany in the trees). BavPat1 sequence was part of a B local clade described in Bavaria, resulting from a single travel-associated Chinese primary case (Rothe C et al., NEJM 2020), and pointed as the initial outbreak favoring the Italian one. None of the 346 Italian sequences analysed with exception for 1 belonged to lineage B, and the only one B sequence found in our dataset differed to BavPat1 for 6 SNP. The lack of a local outbreak involving BavPat1 suggested that the initial outbreak in Bavaria was unlikely to be responsible for directly seeding Lombardy, accordingly with recent published evidences (Worobei M et al, Biorxiv 2020).

Lineage B.1 was initially detected in Lombardy in the second half of February, and in the early of March was present in the Netherlands, UK, Central Europe and USA (Supplementary Figure 4 and Figure 5). Viral sequences were characterized by SNPs at positions 14408 (C to T, non-syn P to L in RdRp) and 23403 (C to T, non-syn D to G in S), with an intra-patient prevalence >95%.

Lineage B.1 was characterized by a sustained local transmission, with no evidences of foreign origins, as suggested by Cluster A. This cluster was characterized by a posterior probability of 0.99 and a tMRCA dated Feb, 14 (Feb, 5 - Feb, 20). It was almost exclusively composed by sequences from Lombardy, with the exception of a Swiss 25 years old patient diagnosed on February 27, a sequence obtained from a Lithuanian patient, whose collection date is partially reported in GISAID (2020-Feb), and a 75 years old Italian patient diagnosed on March 3 in the Italian Marche region. All Cluster A sequences were characterized by the SNP 26530 (A to G, non-syn D to G in M protein; intra-patient prevalence >90%). The relatively small number of foreign sequences belonging to this cluster, and the strain at the bases of this transmission chain (a sequence collected in Bergamo on March 01), let suppose, therefore, that the transmission chain is started in Lombardy. None of patients involved in this cluster reported international nor national travels. Two of them (both from Como) asserted to have been in contact with a confirmed COVID-19 case, thus confirming the evidence of a local clade.

By taking into account the timing of diagnosis, the 40% of the North Italian sequences involved in Cluster A were collected before National lockdown, thus in the early-targeted testing, and the 60% in the later-targeted testing (Table 2). The high-risk area of Bergamo allowed to seed this clade in the time-frame preceding the lockdown. On the contrary, Milan, where the perception of infection risk was probably lower and the mobility was promoted until March, was mainly involved at a later time-point, after the National lock-down.

Overall, these data suggest that a potential origin of lineage B.1 was in Lombardy.

Lineage B.1.1 is a clear descendant of lineage B.1 (posterior probability=1.00) and was simultaneously detected in the late of February in Denmark, Germany, UK, United States, and Lombardy (Figure 5 and Supplementary Figure 4). Two SNPs in N (28881-28883:GGG to AAC, intra-patient prevalence >99%) further characterized the 126 sequences that belonged to this lineage. The exact origin of the entire B.1.1 lineage remains difficult to assess, in the light of simultaneous circulation of this lineage in such a vast territory. Two transmission chains probably starting on late February/early March were detected inside this lineage (Cluster C and D, Figure 5, Supplementary Figure 4, and Table 2). Clusters C and D contained sequences characterized by the non-syn SNP A8072G in NSP3 (N to D), and the syn T4579A in pp1a plus the non-syn G18898T in NSP14 (V to F), respectively. One sequence collected in Como in the first days of March is at the bases of Cluster C that included also two strains isolated in Belgium thereafter. A sequence diagnosed in Atlanta on February, 29 carrying the mutation G18898T but not T4579A, is

at the origin of Cluster D. Unfortunately, no information regarding recent travel history nor contacts are available for these GISAID sequences, neither patients involved in these clusters reported international travels (with exception for one in Cluster C travelling to unknown destination), getting difficult to reconstruct the initial contact tracing contributing to the definition of this clade.

Overall, the simultaneous circulation of lineage B.1.1 in different Countries and the presence of transmission chains with probable foreign origin support the hypothesis that this lineage can have seeded Lombardy by different introductions from geographically distinct sources.

Thirty-four sequences from Lombardy belonged to lineage B.1.5. This lineage can be divided in two main chains, one of them supported by a posterior probability ≥ 0.98 (Cluster B). This cluster described a complex tracing network of 24 persons, who were infected in Lombardy (mainly in Pavia), and diagnosed after March, 8. All sequences were characterized by the SNP 23575 (C to T, non-syn T to I in S protein; inpatient prevalence: $>99\%$). The earliest and most closely related strain was a sequence from China collected on January 24 in the province of Zhejiang. In line with this, the most recent ancestor of this cluster dates back to January 22 (Jan, 18 - Jan, 23), while the transmission chain probably started on Feb, 12 (Feb, 6 - Feb, 23) (Figure 5, Supplementary Figure 4 and Table 2).

The other chain was composed by a total of 26 sequences (11 from Lombardy), all of them characterized by the SNP at position 20268, A to G, syn (nsp15, intra-patient prevalence: 50.0). Sequences from Lombardy were intermixed with sequences from Central and East Europe (n=10), East and West Asia (n=2), North America (n=1), UK and The Netherlands (n=2). Looking at the topology of the trees these sequences seem to be closely related with a subgroup of B.1 sequences collected in the late February in Lombardy (eg. Pavia, Feb 23, and Cremona, Feb 24 even if the support value is <0.10). The low posterior probability and the absence of information regarding recent travel history makes hard to understand if a common origin really exists or their clustering is due to multiple introductions of genetically similar viruses from geographically distinct sources.

These two additional sub-lineages were detected in one and four individuals, respectively. The patient infected by lineage B.1.8 was 84 years old male with no travel history, diagnosed on March, 5 from Lodi. Its infection was preceded from another detected in The Netherlands in March, 04 (Figure 5 and Supplementary Figure 4, posterior probability=0.99). This lineage probably originated by B.1 sequences detected in Lombardy in the late of February (Figure 5 and Supplementary Figure 4). Patients infected by lineage B.1.1.1 come from Milan and Pavia, and did not report any travel history. At the bases of the B.1.1.1 cluster there is a strain collected in UK on March, 13 (Figure 5 and Supplementary Figure 4, posterior probability=1.00), thus suggesting a probable foreign origin of this lineage.

Italy has been the first European country to implement unprecedented measures to restrict individual mobility, and to promote social distancing, with the aim of interrupting transmission of SARS-CoV-2. Following the recognition of the first COVID-19 case in Lombardy, starting from February 21 the government adopted an increasing number of orders, including school and university closures, prohibition of large social gatherings, closure of bar and restaurants, which evolved, ultimately, in a national stay-at-home order.

The strong commercial collaboration between China and the rest of the world implied that, during January, thousands of people (both Chinese and Europeans) travelled from China to Europe and viceversa. In Italy, direct flight from China were blocked on 31 January (while indirect entering in Italy from China through other routes was forbidden only later, see below). However, SARS-CoV-2 infected people have had the change to enter in Italy way before this date, as well proven by the diagnosis of SARS-CoV-2 in Chinese tourists who arrived in Italy on 23 January. In addition, the elimination of direct flights from China, could not prevent the entrance of people though transit flights from other countries (Iacus SM et al, Saf Science 2020). The analysis of the COVID-19 outbreak and the modelling assessment of the effects of travel limitations also confirmed that, despite the strong restrictions on travel to and from China since 23 January 2020, many individuals exposed to SARS-CoV-2 have been traveling internationally without being detected (Chinazzi et al., Science 2020).

Lombardy has 3 international airports (Malpensa, Linate, and Bergamo), and the strongest international commerce activity of all Italy, with a great number of international events organized

constantly in the area. It is thus very likely that a number of infected people, directly from China but also through other countries, entered Lombardy in that period (Goumenou M et al., Mol Med Rep 2020).

The untraced entry of SARS-CoV-2 might be facilitated the diffusion of the virus at population level and its spread locally, as suggested by the epidemiological characteristics of the identified local clusters, mainly sustained by patients with no travel history. Consistently, in the peak of epidemic SARS-CoV-2 diagnosis was mainly addressed to symptomatic cases (prevalently aged people with no travel history) or subjects at high risk of exposure (i.e. health care workers exposed to positive patients without adequate protection). This approach may have caused a substantial underestimation of positive subjects, preventing to include in our analysis asymptomatic infections, whose role in the transmission chains and in influencing the evolution of epidemic remains a challenge to investigate.

These statements are now reported in the revised version of the manuscript (Results, lines 183-290, Discussion, lines 313-339, 394-399).

2. With reference to the above comment, the phylogenetic analysis would benefit from alignment with the proposed global lineage derivation system Pangolin <https://pangolin.cog-uk.io/> (or a suitable equivalent) so that the lineages found can be compared with others on the global tree.

Answer: Lineages of the 346 SARS-CoV-2 sequences included in this manuscript were assigned according to the PANGOLIN application (Pangolin <https://pangolin.cog-uk.io/> and Rambaut A et al., Nature Microb 2020). In order to represent the global diversity of the lineages at May 2020, while minimizing the impact of sampling bias, all whole-genome SARS-CoV-2 sequences available on GISAID at May 3, 2020 (n=3244; GISAID.org), were downloaded and submitted to the Pangolin application. Sequences from GISAID that were error-rich, those without a date of sampling, and identical sequences from each country outbreak were removed. Lastly, the dataset was reduced to 395 sequences by only retaining the earliest, and the most recently sampled sequences from each country outbreak (range of dates: December, 24 2019 – April, 4 2020). Lineages distribution according with collection date and genetic distance were reported in Figure 2 Panel A and B, and in Supplementary data 1, 2 and 3. The topology of Maximum Likelihood and Bayesian Trees obtained by merging the 346 sequences from Lombardy and the 395 GISAID sequences are showed in Figures 4 and 5, and Supplementary Figure 4. The whole manuscript was extensively revised and significantly improved in light of this new classification.

3. A more minor point, it is not discussed how homoplasy is dealt with in the analysis.

Answer: Following reviewer's recommendation, alignment positions showing significant homoplasy were identified by a combined approach, in order to account for regions which might potentially be the result of hypervariability or sequencing artifacts. Homoplasies were firstly identified using HomoplasyFinder, and then confirmed by Treetime (homoplasy setting). In detail, MPBoot was run on the alignment to reconstruct the Maximum Parsimony tree, and to assess branch support by 1000 replicates (-bb 1000). The resulting Maximum Parsimony tree file was used, together with the input alignment, to rapidly identify homoplasies by HomoplasyFinder (Crispell et al., 2019). The top-10 significant homoplastic positions identified by HomoplasyFinder, and confirmed in TreeTime (homoplasy setting) (Isabel S et al., 2020), were masked in the final alignment. This part is now present in the revised version of the manuscript (Methods, Lines 99-102, Supplementary Text, Lines 24-34). Homoplasy results were also reported in the Supplementary Data 4.

4. The D614G mutation is mentioned in the text, but the authors have not carried out an analysis of how this relates to clinical outcome or CT - this would be a useful analysis since they have gone to the trouble to categorise severity of the disease by lineage (I'm not sure that the clinical tool used is the best choice - the ordinal WHO scale might be a better one).

Answer: According with the reviewer's comment, we revised the categorization of the disease severity by using the WHO scale, (WHO, Clinical Management of COVID-19, Interim Guidance May 2020, available at <https://apps.who.int/iris/rest/bitstreams/1278777/retrieve>). The severity of the disease was thus classified as: i) mild: symptomatic patients meeting the case definition for COVID-19 without evidence of viral pneumonia or hypoxia; ii) moderate: clinical signs of pneumonia (fever, cough, dyspnoea, fast breathing) but no signs of severe pneumonia, including SpO₂ ≥ 90% on room air; iii) severe: clinical signs of pneumonia (fever, cough, dyspnoea, fast breathing) plus one of the following: respiratory rate > 30 breaths/min; severe respiratory distress; or SpO₂ < 90% on room air; iv) critical: presence of Acute respiratory distress syndrome (ARDS), and/or sepsis or multiorgan failure (septic shock). This is now reported in the new version of the manuscript (Methods, lines 55-61). The revised classification is also reported in the new version of Table 1.

Regarding the mutation D614G (23403, A to G, non-syn D to G in S), it was present in the 99.4% (n=344/346) of SARS-CoV-2 sequences from Lombardy (all with an intra-patient prevalence >99%), and not as erroneously reported in the first version of the manuscript in 67.8% of sequences. Table 1 and Figure 2A were revised according to this evidence. Among the 237 patients for those disease severity classification was available, 235 were infected by a SARS-CoV-2 carrying the D614G. Critical, severe, moderate and mild manifestations were present in 4.2% (n=10), 22.1% (n=52), 22.1% (n=52), and 50.2% (n=118) of patients carrying the D614G. Median (IQR) Ct value was 18.8 (16.8-20.1). The two patients infected by a wild-type virus at position 614 were characterized by a mild and severe COVID-19 manifestation, respectively. Their Ct values were 16.3 and 20.1, respectively.

D614G variant was recently associated with lower Ct values, indicative of potentially higher in vivo viral loads (Korber et al., Cell 2020). The presence of D614G in almost all the sequences sampled in Lombardy suggests the dominance of this mutation since the first phases of epidemic in Lombardy, and in Europe, confirming its rapid spread and persistence. Unfortunately, the fixation of this mutation in our region prevented us to define any association with disease status and Ct values. However, the quite low Ct values observed in our population (median, IQR: 18.8 16.8-20.1) could be in line with the association of D614G variant with lower Ct values, indicative of potentially higher in vivo viral loads, (Korber et al., Cell 2020), and thus related to severe outcome in COVID-19 (Pujadas et al., Lancet Respir Med 2020; Magleby, et al. Clin Infect Dis. 2020; Alteri C, Cento V, et al Clin Infect Dis 2020). This hypothesis is now discussed in the new version of the manuscript (Discussion, lines 363-378).

5. It was not clear how the samples analysed were selected (was this randomised selection of the total, or governed by CT value or availability?), nor how they relate to the incidence of infection in different parts of Lombardy. Figure 1 shows a nice geographical coverage but would be strengthened by a second panel showing the numbers diagnosed in each area.

Answer: We clarified the sampling criteria of the 371 samples by providing a supplementary chart (Supplementary Figure 2). From February 22 through April 4, 25,082 adult individuals were screened for SARS-CoV-2 infection at two major hospital in Lombardy. 11,445 were tested positive. For 9,251 patients we were able to retrieve information regarding sex, age, and residence. In order to exclude sampling bias that could affect viral diversity, only one patient per family unit was selected (n=7,617). In order to have the measure of viral load of the selected samples, samples with Ct available were retrieved (n=1,561 samples). To warrant high quality sequences and good genomic coverage, samples with Ct values >35 cycles (n=418) were excluded. Out of the remaining 1,143 patients, 371 samples were selected for inclusion, according to the

geographical distribution of COVID-19 cases. In a new Supplementary Table 1, the characteristics of the 9,251 Sars-CoV-2 infected patients with sex, age, and residence information available, were compared with the 371 selected samples. Likelihood Ratio Test, followed by a multinomial logistic regression model to estimate 95% confidence intervals of odds ratios, was used to compare demographic and clinical findings between general and selected SARS-CoV-2 infected populations. By looking at sex, age distribution, the selected population is well representative of SARS-CoV-2 infected general population at that time (now the epidemiology is substantially different). Prevalence of chronic comorbidities is also similar, with the exception of a higher prevalence of cardiovascular and lung diseases in the selected population, compared to the general one (33.2% vs. 24.5%, $P<0.001$; and 14.2% vs. 11.3%, $P=0.04$, respectively). Disease severity and evidence of interstitial pneumonia were largely comparable, even though a lower prevalence of critical COVID-19 cases was observed in the selected population (4.3% vs. 9.6% in the general population; $P=0.001$). The most frequent symptom observed is fever in both populations (66.0% and 63.4%; $P=0.290$), followed by cough and dyspnea, whose prevalence were lower in the selected population (46.0% vs. 52.2% in general population, $P=0.001$; and 38.8% vs. 50.1% in general population, $P<0.001$). The geographical distribution is also comparable between general and selected populations, with the exception of Milan, Pavia and Como. At this regard, it should be noted that this retrospective study involved two major Hospitals localized in Milan and Pavia. Consequently, most of the SARS-CoV-2 infected population resided in these 2 provinces (Milan: 31.1%; Pavia: 25.8%). In order to balance the geographical distribution according with population density and general prevalence of COVID-19 cases, patients from Milan and Pavia were under-sampled down to 20.6% and 19.2% of the selected population, respectively. A higher prevalence of patients residing in Como remained in the selected population respect to general ones (19.2% vs. 8.7%, $P<0.001$).

In order to better represent the geographical distribution of our samples, in relation with COVID-19 prevalence and population density in Lombardy, a new figure 1 was provided. This figure reports both the geographic distribution of COVID-19 confirmed cases and population density among the 12 provinces of Lombardy (Panel A), and the geographic distribution of the 346 SARS-CoV-2 genomes within the lineages detected (Panel B).

It should be noted that, even though the number of samples here analysed is noteworthy, the SARS-CoV-2 epidemic in the East (i.e. Brescia and Mantua) and the valleys of the North (i.e. Valtellina, and Valcamonica) is less represented (Figure 1). Yet, with the sole exception of Brescia (that resembled Bergamo, Cremona and Lodi for SARS-CoV-2 spread), Mantua and the valleys today account only for the 6.7% of confirmed COVID-19 cases in Lombardy, a percentage that does not represent a major issue for the good representability of SARS-CoV-2 epidemic in the region.

These statements are now reported in the revised version of the manuscript (Results, lines 123-126, Supplementary Text, lines 79-110, Discussion, lines 400-409, Supplementary Figure 2 and Supplementary Table 1).

6. I would have been interested to see the sequence data - the authors plan to upload to GISAID - would be useful if this could be uploaded (useful also to the community in general).

Answer: Sequence data have been deposited in GISAID portal and European Nucleotide Archive under the accession numbers EPI_ISL_542098-EPI_ISL_542443 and ERX4545164-ERX4548732, respectively.

Reviewer #2

Remarks to the Author:

This study on the genomic epidemiology of SARS-CoV-2 in Italy attempts to uncover the introduction and transmission of the virus in Lombardy from February to April 2020. Lombardy was one of the hardest hit regions within Italy and was a potential hotspot for transmission to the rest of Europe. At a first glance this study is potentially interesting as it examines some of the earliest SARS-CoV-2 infections from Italy and thereby from a phylogenetic perspective it could fill some missing gaps in the early course of the pandemic within Europe. However, there are many issues that detract from the potential of this study and some of the conclusions are not fully justified.

1. One of my major concerns is the fact that it is not clear that these sequences actually currently exist on GISAID. In fact, a quick look at GISAID reveals that this sequence data is not currently uploaded. While this is the author's prerogative, I find it somewhat troubling that the authors have not shared this data given its potential importance during a pandemic crisis.

Answer: Sequence data have been deposited in the GISAID portal and European Nucleotide under the accession numbers EPI_ISL_542098-EPI_ISL_542443 and ERX4545164-ERX4548732, respectively.

2. This aside from the idea that there are multiple introductions derived from two major lineages is not overly novel and I wonder why the authors have not expanded on this to suggest where the introductions were derived from. Most studies to date have shown that the virus was introduced on multiple independent occasions so why would Italy be any different. Also, in the absence of sequence data from China added to the phylogenetic trees I would suggest that stating two introductions is being very conservative as it is more likely that there have been many more independent introductions from China and mainland Europe.

Answer: As noted by the reviewer, the multiple and dislocated introductions of SARS-CoV-2 in Lombardy are in line with recent descriptions of SARS-CoV-2 spread dynamics in densely populated areas, like New York State, Belgium, and (more recently) The Netherlands (Gonzalez-Reiche AS et al., *Science*, 2020; Dellicour S et al., *Biorxiv*, 2020; Munnink OBB et al., *Nat Med* 2020). The same scenario was also observed in lower population density areas, characterized by regular international travel connections, as Iceland (Gudbjartsson DS et al., *NEJM*, 2020) where government managed to contain the spread of the virus thanks to a strategy of aggressive testing, contact tracing and quarantine. Our data also allow accurate interpretation of the COVID-19 outbreak in Lombardy and its transmission dynamic thanks to a revised phylogenetic approach, although some national-level data have been published (with limited sampling and details) (Lai et al., *Viruses* 2020). In particular, this study, submitted and accepted while our manuscript was under revision, described the circulation of SARS-CoV-2 in the country (characterized by an area of 300,000 km², and a population of 60 million) by analysing 59 whole-genome sequences. Sequences belonged mainly to lineage B.1 (n=47), followed by B.1.1 (n=11) and B.1.5 (n=2). No

local cluster was identified, probably because of the poor representativeness of the study population. In our manuscript, we focused the attention on a territory of 24,000 km², populated by 10 million of people, which accounted for 37% of COVID-19 cases and 53% of deaths of the whole country. Lombardy is also the first territory to be affected by COVID-19 outbreak, with a rate of 112.9 deaths per 100,000 population, almost six times higher than in the rest of Italy. The first phase of epidemic was also characterized by the emergence of many cases concentrated within an extremely short period of time, a scenario that did not grant for the simultaneous and massive circulations of different lineages. The effort to sequence and implement with epidemiological data 346 SARS-CoV-2 sequences belonged to this territory has allowed to describe the heterogeneity of the virus circulating in Lombardy, to appreciate local sustained transmissions and to estimate their probable introduction. At this regard, in the revised version of the manuscript, particular attention was deserved to describe where the introductions were likely derived from, and the potential introduction events that established the transmission chains observed in the region (refers to Results, lines 183-290, Discussion, lines 297-353, Supplementary Figure 4 and Figure 5, Table 2 and points 10 and 12 of this revision).

Finally, the dataset used in the previous version of the manuscript included 50 sequences from China. In the revised version of the manuscript, sequences from each country outbreak (n=395, including 55 from China) were sampled accordingly with lineage and date of sampling (refer to Supplementary Text, lines 35-48). The approach allows to identify more independent introductions from Europe but also from China, as reported at points 3 and 5 of this revision.

3. Greater clarity and investigation should be performed on the role that Italy played in the early establishment of SARS-CoV-2 in Europe as there is still debate on whether an initial outbreak in Bavaria, Germany seeded the Italian outbreak. (See <https://www.biorxiv.org/content/10.1101/2020.05.21.109322v1> for a full discussion). I find it odd that the authors had not discussed this in any great detail.

Answer: To explore the distribution of SARS-CoV-2 sequences in Lombardy and to shed further light on the geographic origin of the SARS-CoV-2 infections in residents of Lombardy, we performed both an estimated maximum likelihood (ML) phylogeny (Figure 4), and a Bayesian molecular clock analysis (Supplementary Figure 4) of 741 viral sequences (346 from complete viral genomes from Lombardy population and 395 from GISAID). To improve the epidemiological characterization of virus spread and local clusters origin, the available information regarding recent travel history or contacts were retrieved from sequences collected in Lombardy (available information for 237 individuals; now present in the revised Table 1 and 2) and GISAID. The information regarding location and recent travel history of the most informative sequences for virus spread and clustering were incorporated in a Bayesian tree interference (Lemey P et al., bioRxiv 2020), showed in Figure 5, in order to yield more realistic reconstructions of virus spread.

According with these revised and additional analyses, the revised version of the manuscript now includes revised versions of Figure 4 (a Maximum Likelihood tree where the lineages distribution is shown), and Supplementary Figure 4 (a Bayesian Tree where the lineages distribution, their most likely origin, and clusters supported by a posterior ≥ 0.98 are shown), a new Figure 5 (a Bayesian Tree incorporating information regarding location and recent travel history) and a new Table 2, reporting the distribution of SARS-CoV-2 local clusters, according to lineages, time of the most recent common ancestors, most closely related strains, timing of diagnosis, age, location and reported risk.

By reassigning sequences to the lineages according to the PANGOLIN application, we found that the majority of sequences (n=179, 51.7%) belonged to lineage B.1, 126 (36.4%) belonged to lineage B.1.1, and 34 (9.8%) were assigned to lineage B.1.5. Lineage B.1.1.1 was found in four North Italian sequences, while A.2, B and B.1.8 lineages were detected in one patient each.

The patient infected by lineage A.2 was diagnosed on March 15 in Lodi. Unfortunately, nor travel information nor possible COVID-19 contact were available for this patient, thus hampering the tracing of the exact dynamic of this infection. Looking at the topology of the trees, the European transmission chain probably originated in Spain (Figure 5 and Supplementary Figure 4) and continued in France in the second half of February. The patient infected by lineage B was a 92 years old male with no travel history. The tree topology suggested that the European transmission chain probably started in UK, and arrived in Italy by untraced individuals (Figure 5 and Supplementary Figure 4).

Most of SARS-CoV-2 sequences from Lombardy (grey taxa with red dots, 344/346 [99.4%]) are interspersed within B sub-lineages (Figure 4).

The molecular clock analysis revealed that the most recent ancestor of these viruses dates back to the second half of January (Jan, 20, 95% HPD intervals: Jan, 16 – Jan, 23, Supplementary Figure 4 and Figure 5), in line with evidences of initial outbreaks in other European countries, in the second half of January (Böhmer MM et al., 2020).

By both ML and Bayesian tree, lineages B.1, B.1.1, B.1.8, B.1.1.1 and B.1.5 seem descendant from a sequence belonged to lineage B and collected on January 28 in Germany (known as BavPat1 and named as Jan, 28 – Germany in the trees). BavPat1 sequence was part of a B local clade described in Bavaria, resulting from a single travel-associated Chinese primary case (Rothe C et al., NEJM 2020), and pointed as the initial outbreak favoring the Italian one. None of the 346 Italian sequences analysed with exception for 1 belonged to lineage B, and the only one B sequence found in our dataset differed to BavPat1 for 6 SNP. The lack of a local outbreak involving BavPat1 suggested that the initial outbreak in Bavaria was unlikely to be responsible for directly seeding Lombardy, accordingly with recent published evidences (Worobei M et al, Biorxiv 2020).

Lineage B.1 was initially detected in Lombardy in the second half of February, and in the early of March was present in the Netherlands, UK, Central Europe and USA (Supplementary Figure 4 and Figure 5). Viral sequences were characterized by SNPs at positions 14408 (C to T, non-syn P to L in RdRp) and 23403 (C to T, non-syn D to G in S), with an intra-patient prevalence >95%.

Lineage B.1 was characterized by a sustained local transmission, with no evidences of foreign origins, as suggested by Cluster A. This cluster was characterized by a posterior probability of 0.99 and a tMRCA dated Feb, 14 (Feb, 5 - Feb, 20). It was almost exclusively composed by sequences from Lombardy, with the exception of a Swiss 25 years old patient diagnosed on February 27, a sequence obtained from a Lithuanian patient, whose collection date is partially reported in GISAID (2020-Feb), and a 75 years old Italian patient diagnosed on March 3 in the Italian Marche region. All Cluster A sequences were characterized by the SNP 26530 (A to G, non-syn D to G in M protein; intra-patient prevalence >90%). The relatively small number of foreign sequences belonging to this cluster, and the strain at the bases of this transmission chain (a sequence collected in Bergamo on March 01), let suppose, therefore, that the transmission chain is started in Lombardy. None of patients involved in this cluster reported international nor national travels. Two of them (both from Como) asserted to have been in contact with a confirmed COVID-19 case, thus confirming the evidence of a local clade.

By taking into account the timing of diagnosis, the 40% of the North Italian sequences involved in Cluster A were collected before National lockdown, thus in the early-targeted testing, and the 60% in the later-targeted testing (Table 2). The high-risk area of Bergamo allowed to seed this clade in the time-frame preceding the lockdown. On the contrary, Milan, where the perception of infection risk was probably lower and the mobility was promoted until March, was mainly involved at a later time-point, after the National lock-down.

Overall, these data suggest that a potential origin of lineage B.1 was in Lombardy.

Lineage B.1.1 is a clear descendant of lineage B.1 (posterior probability=1.00) and was simultaneously detected in the late of February in Denmark, Germany, UK, United States, and Lombardy (Figure 5 and Supplementary Figure 4). Two SNPs in N (28881-28883:GGG to AAC, intra-patient prevalence >99%) further characterized the 126 sequences that belonged to this lineage. The exact origin of the entire B.1.1 lineage remains difficult to assess, in the light of simultaneous circulation of this lineage in such a vast territory. Two transmission chains probably starting on late February/early March were detected inside this lineage (Cluster C and D, Figure 5,

Supplementary Figure 4, and Table 2). Clusters C and D contained sequences characterized by the non-syn SNP A8072G in NSP3 (N to D), and the syn T4579A in pp1a plus the non-syn G18898T in NSP14 (V to F), respectively. One sequence collected in Como in the first days of March is at the bases of Cluster C that included also two strains isolated in Belgium thereafter. A sequence diagnosed in Atlanta on February, 29 carrying the mutation G18898T but not T4579A, is at the origin of Cluster D. Unfortunately, no information regarding recent travel history nor contacts are available for these GISAID sequences, neither patients involved in these clusters reported international travels (with exception for one in Cluster C travelling to unknown destination), getting difficult to reconstruct the initial contact tracing contributing to the definition of this clade.

Overall, the simultaneous circulation of lineage B.1.1 in different Countries and the presence of transmission chains with probable foreign origin support the hypothesis that this lineage can have seeded Lombardy by different introductions from geographically distinct sources.

Thirty-four sequences from Lombardy belonged to lineage B.1.5. This lineage can be divided in two main chains, one of them supported by a posterior probability ≥ 0.98 (Cluster B). This cluster described a complex tracing network of 24 persons, who were infected in Lombardy (mainly in Pavia), and diagnosed after March, 8. All sequences were characterized by the SNP 23575 (C to T, non-syn T to I in S protein; inpatient prevalence: $>99\%$). The earliest and most closely related strain was a sequence from China collected on January 24 in the province of Zhejiang. In line with this, the most recent ancestor of this cluster dates back to January 22 (Jan, 18 - Jan, 23), while the transmission chain probably started on Feb, 12 (Feb, 6 - Feb, 23) (Figure 5, Supplementary Figure 4 and Table 2).

The other chain was composed by a total of 26 sequences (11 from Lombardy), all of them characterized by the SNP at position 20268, A to G, syn (nsp15, intra-patient prevalence: 50.0). Sequences from Lombardy were intermixed with sequences from Central and East Europe ($n=10$), East and West Asia ($n=2$), North America ($n=1$), UK and The Netherlands ($n=2$). Looking at the topology of the trees these sequences seem to be closely related with a subgroup of B.1 sequences collected in the late February in Lombardy (eg. Pavia, Feb 23, and Cremona, Feb 24 even if the support value is <0.10). The low posterior probability and the absence of information regarding recent travel history makes hard to understand if a common origin really exists or their clustering is due to multiple introductions of genetically similar viruses from geographically distinct sources.

These two additional sub-lineages were detected in one and four individuals, respectively. The patient infected by lineage B.1.8 was 84 years old male with no travel history, diagnosed on March, 5 from Lodi. Its infection was proceeded from another detected in The Netherlands in March, 04 (Figure 5 and Supplementary Figure 4, posterior probability=0.99). This lineage probably originated by B.1 sequences detected in Lombardy in the late of February (Figure 5 and Supplementary Figure 4). Patients infected by lineage B.1.1.1 come from Milan and Pavia, and did not report any travel history. At the bases of the B.1.1.1 cluster there is a strain collected in UK on March, 13 (Figure 5 and Supplementary Figure 4, posterior probability=1.00), thus suggesting a probable foreign origin of this lineage.

These statements are now reported in the revised version of the manuscript (Results, lines 183-290, Discussion, lines 297-312).

4. The authors findings that SARS-CoV-2 was present in the region one month before the first diagnosis in Lodi is not proof that it was circulating in the region as it merely reflects the time that the ancestor was circulating somewhere across the globe. The TMRCA is the date of the common ancestor of the sampled genomes in a transmission lineage. While the TMRCA represents the earliest transmission event in the lineage revealed by the data, it does not necessarily represent the first transmission event in the lineage as a whole. Specifically, if the transmission lineage is well sampled then the TMRCA represents the date of the first transmission event in the UK lineage. However, if the transmission lineage is poorly sampled then the TMRCA may represent a later transmission event in the lineage. The actual tMRCA for only Italian sequences is in fact much

later for clusters A and B according to Figure 4. See Figure 1 from <https://virological.org/t/preliminary-analysis-of-sars-cov-2-importation-establishment-of-uk-transmission-lineages/507>

Answer: Thanks to the reviewer's comment, we revised the sentence as follows: BEAST analysis revealed that the most recent ancestor of the viruses circulating in Lombardy dates back to the second half of January (Jan, 20, 95% HPD intervals: Jan, 16 – Jan, 23, Supplementary Figure 4 and Figure 5). In the revised version of the manuscript, particular attention was deserved to describe where the introductions were likely derived from, and which were the most likely events that established the transmission chains in our region (refers to Results, lines 183-290, Discussion, lines 297-312, Supplementary Figure 4 and Figure 5, Table 2 and points 10 and 12 of this revision).

5. The authors state that the sequence data from Figure 3 and 4 does not include those sequences derived from China as they wanted to examine transmission chains. Removing sequence data from China is not the best way to do this as they may be artificial creating transmission chains. If the authors added sequence data from Wuhan and elsewhere in China how do these results change?

Answer: We apologize with the reviewer if the statement reported in the manuscript was not sufficiently clear. With the sentence "These lineages did not contain viral strains isolated in the first months of the outbreak in China (black branches, no dots); this let us hypothesize a transmission chain not directly involving China (i.e., the country where the pandemic originated)" we would state that sequences from China were not directly involved in the origin of the most represented lineages in Lombardy, and not that we excluded sequences from China by phylogenetic analysis. Indeed, the dataset that we used in the previous version of the manuscript, included 50 sequences from China as reported along the manuscript and in Figure 4 and 5. In the new version of the manuscript, sequences from China were 55 and were sampled accordingly with lineage and date of sampling (see Supplementary Data 1 and 2). Again, Chinese sequences (again represented by black taxa without dots in the ML and in the Bayesian Trees) were poorly intermixed with sequences from Lombardy (crf. Figure 5 and Supplementary Figure 4) and did not represented the origin of most local transmission chains, with the exception of the sequence belonged to B.1.5 lineage and collected in January 24 in in the province of Zhejiang (GISAID Code:EPI_ISL_422425), representing one of the earliest and most closely related strains for Cluster B (blue branches in ML and Bayesian Trees). Results and Discussion section of the manuscript were revised accordingly (refers to Results, lines 272-271, Discussion, lines 323-339, Supplementary Figure 4 and Figure 5, Table 2)

6. Why did the authors choose metagenomic sampling? What percentage of reads were SARS-CoV-2? Were there other co-infections? If the authors use such approach, they should go into some level of detail on their sequencing findings such as the above.

Answer: We did not use a metagenomic sampling, but virus genomes were generated by using a multiplex approach, using version 1 of the CleanPlex SARS-CoV-2 Research and Surveillance Panel (<https://www.paragongenomics.com/product/cleanplex-sars-cov-2-panel/>), according to the manufacturer's protocol starting with 50 ng total RNA and followed by Illumina sequencing on a NextSeq 500. Briefly, the multiplex PCR was performed with 2 pooled primer mixture and the cDNA reverse transcribed with random primers was used as a template. Reference-based assembly of the raw data was performed as follows. Illumina adapters were removed, and reads were filtered for quality (average q28 threshold and read length > 135 nt) using FASTP. First and last 15 nucleotides were then removed from all reads. The mapping of cleaned reads was

performed against the GenBank reference genome NC_045512.2 (Wuhan, collection date: December 2019) using BWA-mem. Single nucleotide polymorphisms (SNP variants) were called through a pipeline based on samtools/bcftools, and all SNPs having a minimum supporting read frequency of 40% with a depth ≥ 50 were retained. Consensus were generated using the github freely distributed software vcf_consensus_builder (https://github.com/peterk87/vcf_consensus_builder). Reads mapping on SARS-CoV-2 reference genome were a median of 99.7% (IQR: 99.5%-99.8%), and were able to cover from 94.0% to 99.7% of the SARS-CoV-2 reference genome (GenBank: NC_045512.2), independently from SARS-CoV-2 load (Supplementary Figure 3 A and B). The few genome regions (N=4) with lower reads coverage were consistently limited to no more than 35 nt (Supplementary Figure 3C). Source data for Supplementary Figure 3 were also submitted with the revised version of the manuscript. This is now better specified in the revised version of the manuscript (Methods, lines 62-91).

7. Lineages and sequences should be genotyped according to the PANGOLIN application by Rambaut et al. (<https://www.biorxiv.org/content/10.1101/2020.04.17.046086v1>) as this is becoming the universal classification scheme and it will help orientate the readers.

Answer: This point has been raised also by the reviewer 1 (point 2). The following is the answer there reported:

Lineages of the 346 SARS-CoV-2 sequences included in this manuscript were assigned according to the PANGOLIN application (Pangolin <https://pangolin.cog-uk.io/> and Rambaut A et al., Nature Microb 2020). In order to represent the global diversity of the lineages at May 2020, while minimizing the impact of sampling bias, all whole-genome SARS-CoV-2 sequences available on GISAID at May 3, 2020 (n=3244; GISAID.org), were downloaded and submitted to the Pangolin application. Sequences from GISAID that were error-rich, those without a date of sampling, and identical sequences from each country outbreak were removed. Lastly, the dataset was reduced to 395 sequences by only retaining the earliest, and the most recently sampled sequences from each country outbreak (range of dates: December, 24 2019 – April, 4 2020). Lineages distribution according with collection date and genetic distance were reported in Figure 2 Panel A and B, and in Supplementary data 1, 2 and 3. The topology of Maximum Likelihood and Bayesian Trees obtained by merging the 346 sequences from Lombardy and the 395 GISAID sequences are showed in Figures 4 and 5, and Supplementary Figure 4. The whole manuscript was extensively revised and significantly improved in light of this new classification.

8. Sampling bias and variation. What sampling criteria and justification was used for selecting 371 samples from 11,445 COVID-19 positive specimens? This only represents 3% of the positive population so I am concerned why the authors did not choose more samples and are this small subset representative of the age distribution? Table 1 could be expanded to look at the characteristics of these 11,445 and statistics used to evaluate whether there was any significant difference between those sequenced and those not sequenced.

Answer: We agree with the reviewer on the necessity to further clarify our inclusion criteria. With such purpose, we included further supplementary information in the revised version of the manuscript (Supplementary Table 1, Supplementary Figure 2, Supplementary Text, lines 79-110).

In detail, from February 22 through April 4, 25,082 adult individuals were screened for SARS-CoV-2 infection at two major hospital in Lombardy. 11,445 were tested positive. For 9,251 patients we were able to retrieve information regarding sex, age, and residence. In order to exclude sampling bias that could affect viral diversity, only one patient per family unit was selected (n=7,617). In

order to have the measure of viral load of the selected samples, samples with Ct available were retrieved (n=1,561 samples). To warrant high quality sequences and good genomic coverage, samples with Ct values >35 cycles (n=418) were excluded. Out of the remaining 1,143 patients, 371 samples were selected for inclusion, according to the geographical distribution of COVID-19 cases. In a new Supplementary Table 1, the characteristics of the 9,251 Sars-CoV-2 infected patients with sex, age, and residence information available, were compared with the 371 selected samples. Likelihood Ratio Test, followed by a multinomial logistic regression model to estimate 95% confidence intervals of odds ratios, was used to compare demographic and clinical findings between general and selected SARS-CoV-2 infected populations. By looking at sex, age distribution, the selected population is well representative of SARS-CoV-2 infected general population at that time (now the epidemiology is substantially different). Prevalence of chronic comorbidities is also similar, with the exception of a higher prevalence of cardiovascular and lung diseases in the selected population, compared to the general one (33.2% vs. 24.5%, $P<0.001$; and 14.2% vs. 11.3%, $P=0.04$, respectively). Disease severity and evidence of interstitial pneumonia were largely comparable, even though a lower prevalence of critical COVID-19 cases was observed in the selected population (4.3% vs. 9.6% in the general population; $P=0.001$). The most frequent symptom observed is fever in both populations (66.0% and 63.4%; $P=0.290$), followed by cough and dyspnea, whose prevalence were lower in the selected population (46.0% vs. 52.2% in general population, $P=0.001$; and 38.8% vs. 50.1% in general population, $P<0.001$). The geographical distribution is also comparable between general and selected populations, with the exception of Milan, Pavia and Como. At this regard, it should be noted that this retrospective study involved two major Hospitals localized in Milan and Pavia. Consequently, most of the SARS-CoV-2 infected population resided in these 2 provinces (Milan: 31.1%; Pavia: 25.8%). In order to balance the geographical distribution according with population density and general prevalence of COVID-19 cases, patients from Milan and Pavia were under-sampled down to 20.6% and 19.2% of the selected population, respectively. A higher prevalence of patients residing in Como remained in the selected population respect to general ones (19.2% vs. 8.7%, $P<0.001$).

In order to better represent the geographical distribution of our samples, in relation with COVID-19 prevalence and population density in Lombardy, a new figure 1 was provided. This figure reports both the geographic distribution of COVID-19 confirmed cases and population density among the 12 provinces of Lombardy (Panel A), and the geographic distribution of the 346 SARS-CoV-2 genomes within the lineages detected (Panel B). It should be noted that, even though the number of samples here analysed is noteworthy, the SARS-CoV-2 epidemic in the East (i.e. Brescia and Mantua) and the valleys of the North (i.e. Valtellina, and Valcamonica) is less represented (Figure 1). Yet, with the sole exception of Brescia (that resembled Bergamo, Cremona and Lodi for SARS-CoV-2 spread), Mantua and the valleys today account only for the 6.7% of confirmed COVID-19 cases in Lombardy, a percentage that does not represent a major issue for the good representability of SARS-CoV-2 epidemic in the region (Main manuscript, Discussion, lines 400-409).

9. Similarly, as stated above any statistical differences in the p-values between lineages 1 and 2 could simply reflect a sampling choice and not truly reflect the dynamics of the virus.

Answer: Before specifically addressing this comment, it should be noted that Table 1 and Figure 1 were revised according to the new lineages' assignment (Rambaut et al. Nat Microb, 2020).

By looking at these revised materials, we can observe the presence of three major lineages in Lombardy (B.1, B.1.1, and B.1.5) characterized (as observed in the previous version) by some differences in their geographical distribution. Lineage B.1, probably characterized by an origin in Lombardy, mostly interested the South of this Region, including Lodi, Cremona and Mantua, as accounted for the 90.2% of sequences coming from these provinces (55/61, Figure 1 and Table 1). These territories represent less densely populated area, with modest links to Europe and outside, making plausible a modest lineage diversification. Lineage B.1.1 predominated in the North of

Lombardy, mostly in Como, Lecco, Bergamo and its adjacent territories (such as Alzano and Nembro), and accounted for the 62.7% of sequences coming from these provinces (84/134, Figure 1 and Table 1). These territories are densely populated area, inserted within dense motorway and railway networks, and served by 2 international airports. Thus, we cannot exclude a foreign origin of this lineage, as indeed supported by the characteristics of the local clusters identified inside it. Lastly, lineage B.1.5 was mainly detected in Pavia, because of a local cluster detected in this area and involved 23 B.1.5 sequences. Of note, Milan, the most important urban area of Lombardy, represents the only province in which all lineages are equal represented ($P=0.108$). Given the central location of Milan and its strong economical and transportation links to Europe and outside, it seems plausible the simultaneous circulation of the different lineages in this area.

Among the limitations of this manuscript, we have to take into account that in the peak of epidemic SARS-CoV-2 diagnosis was mainly addressed to symptomatic cases (prevalently aged people with no travel history) or subjects at high risk of exposure (i.e. health care workers exposed to positive patients without adequate protection). This approach may have caused a substantial underestimation of positive subjects, preventing to include in our analysis asymptomatic infections, whose role in the transmission chains and in influencing the evolution of epidemic remains a challenge to investigate.

Overall, even if the sampling scheme used in this study (please refer to point 8 of this revision) suggests that the probability that a bias exists is likely low, we cannot exclude that denser sampling (both temporally and spatially) could reveal novel dispersal patterns and transmission chains not observed here.

All this part is reported in the revised version of the manuscript (Discussion, lines 292-312 and 394-399).

10. Is there any epi data to support these clusters A and B. Is it not conceivable that infected individuals for instance in cluster B were all infected elsewhere (outside Italy) from a common source and then these cases were imported giving the false impression of a Italian cluster.

Answer: As deeply described at the point 3 of this revision, to improve the epidemiological characterization of virus spread and local clusters origin, the available information regarding recent travel history or contacts were retrieved from sequences collected in Lombardy (available information for 237 individuals; now present in the revised Table 1 and 2) and GISAID. The information regarding location and recent travel history of the most informative sequences for virus spread and clustering were incorporated in a Bayesian tree interference (Lemey P et al., bioRxiv 2020), showed in Figure 5, in order to yield more realistic reconstructions of virus spread.

By reassigning lineages of the 346 SARS-CoV-2 sequences included in this manuscript according to the PANGOLIN application by Rambaut et al. (Pangolin <https://pangolin.cog-uk.io/> and Rambaut A et al., Nature Microb 2020), ex-cluster A resulted to account for lineage B.1, while ex-cluster B for lineage B.1.5.

Ex-cluster B was composed by a total of 26 sequences (11 from Lombardy), all of them characterized by the SNP at position 20268, A to G, syn (nsp15, intra-patient prevalence: 50.0). Sequences from Lombardy were intermixed with sequences from Central and East Europe (n=10), East and West Asia (n=2), North America (n=1), UK and The Netherlands (n=2). Looking at the topology of the trees these sequences seem to be closely related with a subgroup of B.1 sequences collected in the late February in Lombardy (eg. Pavia, Feb 23, and Cremona, Feb 24 even if the support value is <0.10). The low posterior probability and the absence of information regarding recent travel history makes hard to understand if a common origin really exists or their clustering is due to multiple introductions of genetically similar viruses from geographically distinct sources. For this reason, it is no longer present in Table 1 and Table 2, yet discussed in the manuscript (Results lines 272-280).

Ex-cluster A (named again cluster A in the new version of the manuscript) was characterized by a posterior probability of 0.99 and a tMRCA dated Feb, 14 (Feb, 5 - Feb, 20). It was almost exclusively composed by sequences from Lombardy, with the exception of a Swiss 25 years old patient diagnosed on February 27, a sequence obtained from a Lithuanian patient, whose collection date is partially reported in GISAID (2020-Feb), and a 75 years old Italian patient diagnosed on March 3 in the Italian Marche region. All Cluster A sequences were characterized by the SNP 26530 (A to G, non-syn D to G in M protein; intra-patient prevalence >90%). The relatively small number of foreign sequences belonging to this cluster, and the strain at the bases of this transmission chain (a sequence collected in Bergamo on March 01), let suppose, therefore, that the transmission chain is started in Lombardy. None of patients involved in this cluster reported international nor national travels. Two of them (both from Como) asserted to have been in contact with a confirmed COVID-19 case, thus confirming the evidence of a local clade.

By taking into account the timing of diagnosis, the 40% of the North Italian sequences involved in Cluster A were collected before National lockdown, thus in the early-targeted testing, and the 60% in the later-targeted testing (Table 2). The high-risk area of Bergamo allowed to seed this clade in the time-frame preceding the lockdown. On the contrary, Milan, where the perception of infection risk was probably lower and the mobility was promoted until March, was mainly involved at a later time-point, after the National lock-down.

By revising the manuscript in order to highlight local transmission chains, we found three additional clusters (2 belonged to lineage B.1.1 and 1 belonged to lineage B.1.5) supported by a posterior probability ≥ 0.98 . Refer to Results, lines 183-290, Figure 5, Supplementary Figure 4, Table 2, of the revised version of the manuscript for a detailed description of these clades.

11. Why did the authors use the “best-fit model of nucleotide substitution GTR+I8 with gamma-distributed rate” for ML tree reconstruction but HKY+Q4 was used in BEAST? Why was not the same model implemented in BEAST. This is confusing and should be clarified. Xml’s file (minus the sequences should be provided).

Answer: This was a mistake in the previous version (thanks for noting) that has been now corrected in the new version of the manuscript. The best-fit model of nucleotide substitution predicted by modeltest-ng-0.1.5 was used for both ML tree and Bayesian Trees. Moreover, according with the reviewers’ recommendations to account for homoplasy and lineages classification (points 7 and 14 of this revision), the final alignment was composed by ten masked nucleotide positions and 738 sequences (GISAID: 395; Lombardy:346). This caused a different prediction of the best-fit model, that changes from a GTR+I with gamma-distributed rate variation to a GTR+I. Methods section was revised accordingly (see Manuscript, Methods, lines 103-113, Supplementary Text, lines 49-77). The Beast xml files are also provided (see Manuscript, Code availability section).

12. How are the authors sure that the measured diversity illustrated in the supplementary figures is not measuring re-introduction of the virus from travel cases?

Answer: After lineage reassignment by PANGOLIN, Supplementary Figure 3 was extensively revised and moved in the main text as Figure 2. To improve the epidemiological characterization of our results, all available information on travel history or COVID-19 contacts were retrieved for 237 individuals and reported in the revised versions of Table 1 and Table 2. Only 4 patients out of 237 declared an international travel history. For three of them the destination was not specified, while one individual, reported a travel in Singapore from January 12 to February 5, one month before to be found infected by a B.1.1 strain (March 10). For epidemiological insights on lineages and clusters, please refer to point 1 of reviewer 1 and at points 3 and 10 of this reviewer and to revised version of the manuscript (Results, lines 145-182, 183-290, Discussion, lines 298-312).

13. The manuscript is devoid of some level of detail on their global dataset. GISAID numbers should be given as a supplementary table to illustrate the breakdown of sequences per country. As the phylogenetic results may be somewhat sensitive to how these sequences were selected.

Answer: The list of available whole-genome SARS-CoV-2 sequences (n = 3244) on GISAID (GISAID.org) on 3 May 2020 were present in Supplementary data 1. The list of the 392 GISAID sequences selected is reported as Supplementary data 2. For further detail on study-population's selection please refer to point 7 of this revision.

14. Where any sites in the alignment masked for issues related to homoplastic substitutions (see <https://virological.org/t/issues-with-sars-cov-2-sequencing-data/473>).

Answer: This point has been raised also by the reviewer 1 (point 3). The following is the answer there reported:

Alignment positions showing significant homoplasmy were identified by a combined approach, in order to account for regions which might potentially be the result of hypervariability or sequencing artifacts. Homoplasies were firstly identified using HomoplasmyFinder, and then confirmed by Treetime (homoplasmy setting). In detail, MPBoot was run on the alignment to reconstruct the Maximum Parsimony tree, and to assess branch support by 1000 replicates (-bb 1000). The resulting Maximum Parsimony tree file was used, together with the input alignment, to rapidly identify homoplasies by HomoplasmyFinder (Crispell et al., 2019). The top-10 significant homoplastic positions identified by HomoplasmyFinder, and confirmed in TreeTime (homoplasmy setting) (Isabel S et al., 2020), were masked in the final alignment. This part is now present in the revised version of the manuscript (Methods, Lines 99-102, Supplementary Text, Lines 24-34). Homoplasmy results were also reported in the Supplementary Data 4.

15. Based on sequence data it is difficult to equate genotype to phenotype and the suggestion that because lineages have similar evolutionary characteristics then they have similar replication profiles is unfounded and should be removed (line 172).

Answer: According with the reviewer's comment, this statement was removed.

16. Have the authors examined D614G spike mutation that has been speculated to be linked with disease severity?

Answer: This point has been raised also by the reviewer 1 (point 4). The following is the answer there reported:

D614G was present in the 99.4% (n=344/346) of SARS-CoV-2 sequences from Lombardy (all with an intra-patient prevalence >99%), and not as erroneously reported in the first version of the manuscript in 67.8% of sequences. Table 1 and Figure 2A were revised according to this evidence. Among the 237 patients for those disease severity classification was available, 235 were infected by a SARS-CoV-2 carrying the D614G. Critical, severe, moderate and mild manifestations were present in 4.2% (n=10), 22.1% (n=52), 22.1% (n=52), and 50.2% (n=118) of patients carrying the D614G. Median (IQR) Ct value was 18.8 (16.8-20.1). The two patients infected by a wild-type virus at position 614 were characterized by a mild and severe COVID-19 manifestation, respectively. Their Ct values were 16.3 and 20.1, respectively.

D614G variant was recently associated with lower Ct values, indicative of potentially higher in vivo viral loads (Korber et al., Cell 2020). The presence of D614G in almost all the sequences sampled in Lombardy suggests the dominance of this mutation since the first phases of epidemic in

Lombardy, and in Europe, confirming its rapid spread and persistence. Unfortunately, the fixation of this mutation in our region prevented us to define any association with disease status and Ct values. However, the quite low Ct values observed in our population (median, IQR: 18.8 16.8-20.1) could be in line with the association of D614G variant with lower Ct values, indicative of potentially higher in vivo viral loads, (Korber et al., Cell 2020), and thus related to severe outcome in COVID-19 (Pujadas et al., Lancet Respir Med 2020; Magleby, et al. Clin Infect Dis. 2020; Alteri C, Cento V, et al Clin Infect Dis 2020). This hypothesis is now discussed in the new version of the manuscript (Discussion, lines 363-378).

REVIEWER COMMENTS

Reviewer #1 (Remarks to the Author):

The modified manuscript is hugely improved based on the new analyses incorporated by the authors. I particularly liked the new figures and the incorporation of Pangolin lineage assignments. I think this is an important report from the epicentre of the European epidemic.

I have a few very minor further comments:

1. The English is not perfect throughout and I think it would benefit from some editorial improvements just to clarify a small number of areas and make it easier to read.
2. With regard to the above comment, I couldn't understand the meaning of the paragraph on pages 8-9 lines 212-220. I suspect this was a really interesting bit as it refers to the hypothesis that the Bavarian outbreak seeded the Italian one - I think this is disproven based on the data shown - but it's really unclear in the text - I think this is just a translation issue.
3. There is a typo on page 14 line 362 where it is stated that 7 SNPs distinguished the Italian strains while in the abstract and in Table 1, it is stated that this number was 6.
4. For lineage B.1.5, in Figure 5 and elsewhere in the text it is suggested that the most closely related origin sequence was likely derived from Zhejiang, China. However this sequence is not visible in Figure 4 - why not?
5. The authors assertion that theirs is the largest sequencing study at the beginning of the Discussion is not correct and should be modified.
6. While 7 lineages are evident based on Pangolin classification and the authors suggest that these represent likely different international origins, I'm not sure that this is true. I tend to agree with their conclusion that the virus was circulating undetected for some time before detection and I wonder if the origin of at least 1 or 2 of these lineages is in fact Lombardy. Also, while 7 lineages were detected, 3 were very clearly dominant and I would emphasise this in the text.

Reviewer #2 (Remarks to the Author):

The authors have performed a serious re-write of this manuscript and boosted the study with the analysis of specific circulating lineages. This is a much improved study and I congratulate the authors for performing the Bayesian analyses accommodating travel history and for transparency the authors have also included the BEAST xml files. These points strengthen the manuscript and the authors have presented robust answers to my previous concerns and questions. However, the English could be strengthened in some places particularly in the newly added text. Some minor examples below:

1. Abstract. Remove the word "way" on line 11 as it suggests an extended period of time.
 2. Results: Line 155. "interested" appears to be the incorrect word as the authors are referring to the composition of lineages in different regions. Similarly on line 304.
 3. Line 167. Authors should specify the SARS-CoV-2 reference strain used as the relationship between genetic distance will naturally be lower for those strains within the same lineage as the reference.
- Line 195/96: Remove the word "nor".
Line 220: Evidence instead of "evidences"

We would like to thank the reviewers for their time and for appreciating our work in revising the manuscript in accordance with their comments.

The minor refinements requested are summarized in our response below.

Response to Reviewers' Comments

Reviewer #1

Remarks to the Author:

The modified manuscript is hugely improved based on the new analyses incorporated by the authors. I particularly liked the new figures and the incorporation of Pangolin lineage assignments. I think this is an important report from the epicentre of the European epidemic.

Answer: We thank the reviewer for his/her positive comment.

I have a few very minor further comments:

1. The English is not perfect throughout and I think it would benefit from some editorial improvements just to clarify a small number of areas and make it easier to read.

Answer: English language has been edited by a native speaker, resulting in some changes in the structure of various sentences throughout the manuscript.

2. With regard to the above comment, I couldn't understand the meaning of the paragraph on pages 8-9 lines 212-220. I suspect this was a really interesting bit as it refers to the hypothesis that the Bavarian outbreak seeded the Italian one - I think this is disproven based on the data shown - but it's really unclear in the text - I think this is just a translation issue.

Answer: The paragraph was revised as follows: none of the 346 Italian SARS-CoV-2 sequences here analyzed belonged to lineage B, with the exception of one. This sequence differs from BavPat1 for 6 SNP. Thus, according to recent published evidence, the lack of a local cluster involving BavPat1 suggested that the initial outbreak in Bavaria was unlikely to have been responsible for directly seeding Lombardy (Page 6, lines 143-147).

3. There is a typo on page 14 line 362 where it is stated that 7 SNPs distinguished the Italian strains while in the abstract and in Table 1, it is stated that this number was 6.

Answer: We thank the reviewer for noting this typo, now corrected accordingly (Page 11, lines 288-289).

4. For lineage B.1.5, in Figure 5 and elsewhere in the text it is suggested that the most closely related origin sequence was likely derived from Zhejiang, China. However, this sequence is not visible in Figure 4 - why not?

Answer: The sequence collected on January 24 in the province of Zhejiang (named G59|Chi|24|01|2020) is the earliest and most closely related strain of Cluster B, describing a B.1.5 complex tracing network of 24 persons, who were infected in Lombardy (mainly in Pavia), and diagnosed after March, 8. This sequence is present in the trees described in Figure 2 (ex Figure 4), Figure 5 and Supplementary Fig. 4. In all trees this sequence is positioned at the basis of Cluster B (in blue).

5. The authors assertion that theirs is the largest sequencing study at the beginning of the Discussion is not correct and should be modified.

Answer: We modified the sentences as follows: These data on the genomic epidemiology of SARS-CoV-2 in Lombardy, based on a consistent number of whole genome SARS-CoV-2 sequences circulating in a single region, indicate the simultaneous circulation of at least three widely represented lineages of SARS-CoV-2 (B.1, B.1.1 and B.1.5), supported by transmission chains occurring since the first half of February (Page 9, lines 223-227).

6. While 7 lineages are evident based on Pangolin classification and the authors suggest that these represent likely different international origins, I'm not sure that this is true. I tend to agree with their conclusion that the virus was circulating undetected for some time before detection and I wonder if the origin of at least 1 or 2 of these lineages is in fact Lombardy. Also, while 7 lineages were detected, 3 were very clearly dominant and I would emphasise this in the text.

Answer: In agreement with the reviewer's comment, abstract, discussion and conclusion have been revised in order to emphasize the early circulation of lineages in Lombardy and the central role of this region in the European epidemic of SARS-CoV-2.

Reviewer #2

Remarks to the Author:

The authors have performed a serious re-write of this manuscript and boosted the study with the analysis of specific circulating lineages. This is a much improved study and I congratulate the authors for performing the Bayesian analyses accommodating travel history and for transparency the authors have also included the BEAST xml files. These points strengthen the manuscript and the authors have presented robust answers to my previous concerns and questions. However, the English could be strengthened in some places particularly in the newly added text.

Answer: We thank the reviewer for his/her positive comment. English language has been edited by a native speaker, resulting in some changes in the structure of several sentences, throughout the manuscript.

Some minor examples below:

1. Abstract. Remove the word "way" on line 11 as it suggests an extended period of time.

Answer: Done

2. Results: Line 155. "interested" appears to be the incorrect word as the authors are referring to the composition of lineages in different regions. Similarly on line 304.

Answer: Accordingly, "Interested" was replaced by "affected" (Pages 4 and 9, lines 83 and 229).

3. Line 167. Authors should specify the SARS-CoV-2 reference strain used as the relationship between genetic distance will naturally be lower for those strains within the same lineage as the reference.

Answer: We agree with the referee's comment and the sentence was thus revised as follows: Looking at the genetic distance with respect to SARS-CoV-2 reference strain (belonging to lineage B), it was naturally lower for isolates in lineage B.1 with respect to isolates in lineage B.1.1 and B.1.5 (1.4×10^{-4} [1.0×10^{-4} ; 1.7×10^{-4}] vs 1.7×10^{-4} [1.4×10^{-4} ; 1.7×10^{-4}] vs 2.4×10^{-4} [2.1×10^{-4} ; 2.7×10^{-4}], $P < 0.001$, Figure 3b), thus indicating a closer genetic relatedness to original B strains for lineage B.1 with respect to the others (Page 4, lines 93-97).

4. Line 195/96: Remove the word "nor".

Answer: This sentence has been modified as follows: travel information and possible COVID-19 contact were unavailable for this patient, thus hampering the tracing of the exact dynamic of the infection. (Page 5, lines 123-124)

5. Line 220: Evidence instead of "evidences"

Answer: Done